# Common Foundations for SHACL, ShEx, and PG-Schema

## ABSTRACT

Graphs have emerged as an important foundation for a variety of applications, including capturing and reasoning over factual knowledge, semantic data integration, social networks, and providing factual knowledge for machine learning algorithms. To formalise certain properties of the data and to ensure data quality, there is a need to describe the *schema* of such graphs. Because of the breadth of applications and availability of different data models, such as RDF and property graphs, both the Semantic Web and the database community have independently developed *graph schema languages*: SHACL, ShEx, and PG-Schema. Each language has its unique approach to defining constraints and validating graph data, leaving potential users in the dark about their commonalities and differences. In this paper, we provide formal, concise definitions of the *core components* of each of these schema languages. We employ a uniform framework to facilitate a comprehensive comparison between the languages and identify a common set of functionalities, shedding light on both overlapping and distinctive features of the three languages.

**ACM Reference Format:**
Anonymous Author(s). 2024. Common Foundations for SHACL, ShEx, and PG-Schema. In *Proceedings of The Web Conference.* ACM, New York, NY, USA, 20 pages. https://doi.org/10.1145/nnnnnnn.nnnnnnn

## 1 INTRODUCTION

Driven by the unprecedented growth of interconnected data, *graph-based data representations* have emerged as an expressive and versatile framework for modelling and analysing connections in data sets [46]. This rapid growth however, has led to a proliferation of diverse approaches, each with its own identity and perspective.

The two most prominent graph data models are *RDF* (Resource Description Framework) [14] and *Property Graphs* [9]. In RDF, data is modelled as a collection of triples, each consisting of a subject, predicate, and object. Such triples naturally represent either edges in a directed labelled graph (where the predicates represent relationships between nodes), or attributes-value pairs of nodes. That is, objects can both be entities or atomic (literal) values. In contrast, Property Graphs model data as nodes and edges, where both can have labels and records attached, allowing for a flexible representation of attributes directly on the entities and relationships.

Similarly to the different data models, we are also seeing different approaches towards *schema languages* for graph-structured data. Traditionally, in the Semantic Web community, schema and constraint languages have been *descriptive*, focusing on flexibility to accommodate varying structures. However, there has been a growing need for more *prescriptive* schemas that focus on *validation of data*. At the same time, in the Database community, schemas have traditionally been prescriptive but, since the rise of semi-structured data, the demand for descriptive schemas has been growing. Thus,

the philosophies of schemas in the two communities have been growing closer together.

For RDF, there are two main schema languages: SHACL (Shapes Constraint Language) [29], which is also a W3C recommendation, and ShEx (Shape Expressions) [43]. In the realm of Property Graphs, the current main approach is PG-Schema [2, 3]. The development processes of these languages have been quite different. For SHACL and ShEx, the formal semantics were only introduced after their initial implementations, echoing the evolution of programming languages. Indeed, an analysis of SHACL's expressive power and associated decision problems appeared in the literature [6, 7, 34, 38–40] only after it was published as a W3C recommendation, leading up to a fully *recursive variant* of the language [1, 5, 12, 13, 39], whose semantics had been left undefined in the standard. A similar scenario occurred with ShEx, where formal analyses were only conducted in later phases [8, 48]. PG-Schema developed in the opposite direction. Here, a group of experts from industry and academia first defined the main ideas in a sequence of research papers [2, 3] and the implementation is expected to follow.

Since these three languages have been developed in different communities, in the course of different processes, it is no surprise that they are quite different. SHACL, ShEx, and PG-Schema use an array of diverse approaches for defining how their components work, ranging from *declarative* (formulas that *specify what to look for*) to *generative* (expressions that *generate the matching content*), and even combinations thereof. The bottom line is that we are left with three approaches to express a "schema for graph-structured data" that are very different at first glance.

As a group of authors coming from both the Semantic Web and Database communities, we believe that there is a *need for common understanding*. While the functionalities of schemas and constraints used in the two communities largely overlap, it is a daunting task to understand the essence of languages, such as SHACL, ShEx, and PG-Schema. In this paper, we therefore aim to shed light on the common aspects and the differences between these three languages.

Using a common framework, we provide crisp definitions of the main aspects of the languages. Because the languages operate on different data models, as a first step we introduce the *Common Graph Data Model*, a mathematical representation of data that *canonically embeds* both RDF graphs and Property Graphs (see Section 2, which also develops general common foundations). Precise abstractions of the languages themselves are presented in Sections 3 (SHACL), 4 (ShEx), and 5 (PG-Schema); in the Appendices we explain how and why we sometimes deviate from the original formalisms. Each of these sections contains examples to give readers an immediate intuition about what kinds of conditions each language can express. Then, in Section 6, we present the *Common Graph Schema Language (CoGSL)*, which consists of functionalities shared by them all.

Casting all three languages in a common framework has the immediate advantage that the reader can identify common functionalities *based on the syntax only*: on the one hand, we aim at giving the same semantics to schema language components that

*The Web Conference, 28 April - 2 May 2025, Sydney, Australia*
2024. ACM ISBN 978-x-xxxx-xxxx-x/YY/MM. . . $15.00
https://doi.org/10.1145/nnnnnnn.nnnnnnn

syntactically look the same, and on the other hand, we can provide examples of properties that distinguish the three languages using simple syntactic constructs that are not part of the common core. Aside from corner cases, properties expressed using constructs outside the common core are generally not expressible in all three languages. By providing an understanding of fundamental differences and similarities between the three schema languages, we hope to benefit both practitioners in choosing a schema language fitting their needs, and researchers in studying the complexity and expressiveness of schema languages.

## 2 FOUNDATIONS

In this section we present some material that we will need in the subsequent sections, and define a data model that consists of common aspects of RDF and Property Graphs.

### 2.1 A common data model

When developing a common framework for SHACL, ShEx, and PG-Schema, the first challenge is establishing a *common data model*, since SHACL and ShEx work on RDF, whereas PG-Schema works on Property Graphs. Rather than using a model that generalizes both RDF and Property Graphs, we propose a simple model, called *common graphs*, which we obtained by asking what, fundamentally, are the *common aspects* of RDF and Property Graphs (Appendix A gives more details on the distilling of common graphs).

Let us assume disjoint countable sets of nodes $\mathcal{N}$, values $\mathcal{V}$, predicates $\mathcal{P}$, and keys $\mathcal{K}$ (sometimes called properties).

DEFINITION 1. *A common graph is a pair $\mathcal{G} = (E, \rho)$ where*

- *$E \subseteq_{fin} \mathcal{N} \times \mathcal{P} \times \mathcal{N}$ is its set of edges (which carry predicates), and*
- *$\rho : \mathcal{N} \times \mathcal{K} \rightharpoonup \mathcal{V}$ is a finite-domain partial function mapping node-key pairs to values.*

*The set of nodes of a common graph $\mathcal{G}$, written $\mathrm{Nodes}(\mathcal{G})$, consists of all elements of $\mathcal{N}$ that occur in $E$ or in the domain of $\rho$. Similarly, $\mathrm{Keys}(\mathcal{G})$ is the subset of $\mathcal{K}$ that is used in $\rho$, and $\mathrm{Values}(\mathcal{G})$ is the subset of $\mathcal{V}$ that is used in $\rho$ (that is, the range of $\rho$).*

*Example 1.* Consider Figure 1, containing a graph to store information about *users* who may have access to (possibly multiple) *accounts* in, e.g., a media streaming service. In this example, we have six nodes describing four persons ($u_1, ..., u_4$) and two accounts ($a_1, a_2$). As a common graph $\mathcal{G} = (E, \rho)$, the nodes are $a_1, u_1$, etc. Examples of edges in $E$ are ($u_2$, hasAcccess, $a_1$) and ($u_3$, invited, $u_2$). Furthermore, we have $\rho(u_2, email) = $ d@d.d and $\rho(a_1, card) = 1234$. So, $E$ captures the arrows in the figure (labeled with predicates) and $\rho$ captures the key/value information for each node. Notice that a person may be the owner of an account, and may potentially have access to other accounts. This is captured using the predicates ownsAccount and hasAcccess, respectively. In addition, the system implements an invitation functionality, where users may invite other people to join the platform. The previous invitations are recorded using the predicate invited. Both accounts and users may be privileged, which is stored via a Boolean value of the key *privileged*. We note that the presence of the key *email* (resp. of the key (credit) *card*) is associated with, and indeed identifies users (resp. accounts).

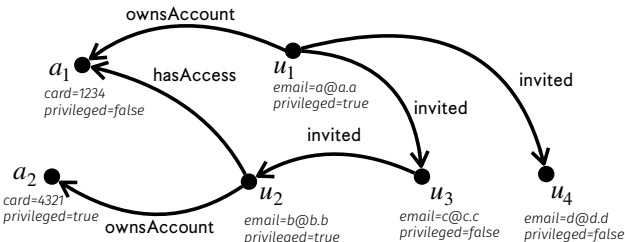

**Figure 1: The media service common graph.**

It is easy to see that a common graph is a special case of a property graph (see [2] for a formal definition of property graphs). A common graph can also be seen as a set of triples, as in RDF. Let

$$\mathcal{E} = (\mathcal{N} \times \mathcal{P} \times \mathcal{N}) \cup (\mathcal{N} \times \mathcal{K} \times \mathcal{V}) .$$

Then, a common graph can be seen as a finite set $\mathcal{G} \subseteq \mathcal{E}$ such that for each $u \in \mathcal{N}$ and $k \in \mathcal{K}$ there is at most one $v \in \mathcal{V}$ such that $(u, k, v) \in \mathcal{G}$. Indeed, a common graph $(E, \rho)$ corresponds to

$$E \cup \{(u, k, v) \mid \rho(u, k) = v\} .$$

When we write $\rho(u, k) = v$ we assume that $\rho$ is defined on $(u, k)$.

*Throughout the paper we see property graph $\mathcal{G}$ simultaneously as a pair $(E, \rho)$ and as a set of triples from $\mathcal{E}$, switching between these perspectives depending on what is most convenient at a given moment.*

### 2.2 Node contents and neighbourhoods

Let $\mathcal{R}$ be the set of all *records*, i.e., finite-domain partial functions $r : \mathcal{K} \rightharpoonup \mathcal{V}$. We write records as sets of pairs $\{(k_1, w_1), \ldots (k_n, w_n)\}$ where $k_1, \ldots, k_n$ are all different, meaning that $k_i$ is mapped to $w_i$.

For a common graph $\mathcal{G} = (E, \rho)$ and node $v$ in $\mathcal{G}$, by a slight abuse of notation we write $\rho(v)$ for the record $\{(k, w) \mid \rho(v, k) = w\}$ that collects all key-value pairs associated with node $v$ in $\mathcal{G}$. We call $\rho(v)$ the *content* of node $v$ in $\mathcal{G}$. This is how PG-Schema interprets common graphs: it views key-value pairs in $\rho(v)$ as *properties* of the node $v$, rather than independent, navigable objects in the graph.

SHACL and ShEx, on the other hand, view common graphs as sets of triples and make little distinction between keys and predicates. The following notion—when applied to a node—uniformly captures the local context of this node from that perspective: the content of the node and all edges incident with the node.

DEFINITION 2 (NEIGHBOURHOOD). *Given a common graph $\mathcal{G}$ and a node or value $v \in \mathcal{N} \cup \mathcal{V}$, the neighbourhood of $v$ in $\mathcal{G}$ is $\mathrm{Neigh}_{\mathcal{G}}(v) = \{(u_1, p, u_2) \in \mathcal{G} \mid u_1 = v \text{ or } u_2 = v\}$.*

That is, when $v \in \mathcal{N}$, then $\mathrm{Neigh}_{\mathcal{G}}(v)$ is a star-shaped graph where only the central node has non-empty content. When $v \in \mathcal{V}$, then $\mathrm{Neigh}_{\mathcal{G}}(v)$ consists of all the nodes in $\mathcal{G}$ that have some key with value $v$, which is a common graph with no edges and a restricted function $\rho$.

### 2.3 Value types

We assume an enumerable set of *value types* $\mathcal{T}$. The reader should think of value types as integer, boolean, date, etc. Formally, for each value type $\mathbb{v} \in \mathcal{T}$, we assume that there is a set $[\![\mathbb{v}]\!] \subseteq \mathcal{V}$ of all values of that type and that each value $v \in \mathcal{V}$ belongs to some

type, i.e., there is at least one $\mathbf{v} \in \mathcal{T}$ such that $v \in [\![\mathbf{v}]\!]$. Finally, we assume that there is a type $\mathtt{any} \in \mathcal{T}$ such that $[\![\mathtt{any}]\!] = \mathcal{V}$.

## 2.4 Shapes and schemas

We formulate all three schema languages using *shapes*, which are unary formulas describing the graph's structure around a *focus* node or a value. Shapes will be expressed in different formalisms, specific to the schema language; for each of these formalisms we will define when a focus node or value $v \in \mathcal{N} \cup \mathcal{V}$ *satisfies* shape $\varphi$ in a common graph $\mathcal{G}$, written $\mathcal{G}, v \models \varphi$.

Inspired by ShEx *shape maps*, we abstract a schema $\mathcal{S}$ as a set of pairs $(sel, \varphi)$, where $\varphi$ is a shape and $sel$ is a *selector*. A selector is also a shape, but usually a very simple one, typically checking the presence of an incident edge with a given predicate, or a property with a given key. A graph $\mathcal{G}$ is *valid wrt.* $\mathcal{S}$, written $\mathcal{G} \models \mathcal{S}$, if

$$\mathcal{G}, v \models sel \quad \text{implies} \quad \mathcal{G}, v \models \varphi$$

for all $v \in \mathcal{N} \cup \mathcal{V}$ and $(sel, \varphi) \in \mathcal{S}$. That is, for each focus node or value satisfying the selector, the graph around it looks as specified by the shape. We call schemas $\mathcal{S}$ and $\mathcal{S}'$ *equivalent* if $\mathcal{G} \models \mathcal{S}$ iff $\mathcal{G} \models \mathcal{S}'$ for all $\mathcal{G}$. In what follows, we may use $sel \Rightarrow \varphi$ to indicate a pair $(sel, \varphi)$ from a schema $\mathcal{S}$.

*Example 2.* We next describe some constraints one may want to express in the domain of Example 1.

(C1) We may want the values associated to certain keys to belong to concrete datatypes, like strings or Boolean values. In our example, we want to state that the value of the key *card* is always an integer.

(C2) We may expect the existence of a value associated to a key, an outgoing edge, or even a complex path for a given source node. For our example, we require that all owners of an account have an email address defined.

(C3) We may want to express database-like uniqueness constraints. For instance, we may wish to ensure that the email address of an account owner uniquely identifies them.

(C4) We may want to ensure that all paths of a certain kind end in nodes with some desired properties. For example, if an account is privileged, then all users that have access to it should also be privileged.

(C5) We may want to put an upper bound on the number of nodes reached from a given node by certain paths. For instance, every user may have access to at most 5 accounts.

## 3 SHACL ON COMMON GRAPHS

We first treat SHACL, because it is conceptually the simplest of the three languages. It is essentially a logic—some call it a *description logic in disguise* [6]. For each of the three languages, we need to perform some minor deviations in order to define it over common graphs. For SHACL, we discuss these in Appendix B. Our abstraction of SHACL on common graphs is inspired by [1, 6, 7, 13, 16].

**Definition 3 (Path Expression).** *A path expression $\pi$ is given by the following grammar:*

$$\pi ::= \mathrm{id} \mid p \mid k \mid \pi^- \mid \pi \cdot \pi \mid \pi \cup \pi \mid \pi^* .$$

*with $p \in \mathcal{P}$, $k \in \mathcal{K}$ and $\mathrm{id}$ the identity relation (or empty word).*

**Table 1: Evaluation of a path expressions.**

| $\pi$ | $[\![\pi]\!]^{\mathcal{G}} \subseteq (\mathcal{N} \cup \mathcal{V}) \times (\mathcal{N} \times \mathcal{V})$ for $\mathcal{G} = (E, \rho)$ |
|---|---|
| id | $\{(v, v) \mid v \in \mathcal{N} \cup \mathcal{V}\}$ |
| $p$ | $\{(v, u) \mid (v, p, u) \in E\}$ |
| $k$ | $\{(v, u) \mid \rho(v, k) = u\}$ |
| $\pi^-$ | $\{(v, u) \mid (u, v) \in [\![\pi]\!]^{\mathcal{G}}\}$ |
| $\pi \cdot \pi'$ | $\{(v, u) \mid \exists v' : (v, v') \in [\![\pi]\!]^{\mathcal{G}} \wedge (v', u) \in [\![\pi']\!]^{\mathcal{G}}\}$ |
| $\pi \cup \pi'$ | $[\![\pi]\!]^{\mathcal{G}} \cup [\![\pi']\!]^{\mathcal{G}}$ |
| $\pi^*$ | $[\![\mathrm{id}]\!]^{\mathcal{G}} \cup [\![\pi]\!]^{\mathcal{G}} \cup [\![\pi \cdot \pi]\!]^{\mathcal{G}} \cup \ldots$ |

**Definition 4 (SHACL shape).** *A SHACL shape $\varphi$ is given by the following grammar:*

$$\varphi ::= \top \mid \mathrm{is}(c) \mid \mathrm{test}(\mathbf{v}) \mid \mathrm{closed}(Q) \mid \mathrm{eq}(\pi, p) \mid$$
$$\mathrm{disj}(\pi, p) \mid \neg\varphi \mid \varphi \wedge \varphi \mid \varphi \vee \varphi \mid \exists^{\geq n}\pi.\varphi \mid \exists^{\leq n}\pi.\varphi .$$

*with $c \in \mathcal{V}$, $\mathbf{v} \in \mathcal{T}$, $Q \subseteq_{fin} \mathcal{P} \cup \mathcal{K}$, $p \in \mathcal{P}$, and $n$ a natural number. We may use $\exists\pi.\varphi$ as syntactic sugar for $\exists^{\geq 1}\pi.\varphi$.*

**Definition 5 (SHACL selector).** *A SHACL selector $sel$ is a SHACL shape of a restricted form, given by the following grammar:*

$$sel ::= \exists p.\top \mid \exists k.\top \mid \exists p^-.\top \mid \exists k^-.\top \mid \mathrm{is}(c) .$$

*with $p \in \mathcal{P}$, $k \in \mathcal{K}$, and $c \in \mathcal{V}$.*

Putting it together, a *SHACL Schema* $\mathcal{S}$ is a finite set of pairs $(sel, \varphi)$, where $sel$ is a SHACL selector and $\varphi$ is a SHACL shape.

To define the semantics of SHACL schemas, we first define in Table 1 the semantics of a SHACL path expression $\pi$ on a graph $\mathcal{G}$ as a binary relation $[\![\pi]\!]^{\mathcal{G}}$ over $\mathcal{N} \cup \mathcal{V}$. The semantics of SHACL shapes is defined in Table 2, which specifies when a node or value $v$ satisfies a SHACL shape $\varphi$ w.r.t. a $\mathcal{G}$, written $\mathcal{G}, v \models \varphi$. Note that both $[\![\pi]\!]^{\mathcal{G}}$ and $\{v \in \mathcal{N} \cup \mathcal{V} \mid \mathcal{G}, v \models \varphi\}$ may be infinite: for example, $[\![\mathrm{id}]\!]^{\mathcal{G}}$ is the identity relation over the infinite set $\mathcal{N} \cup \mathcal{V}$.

The semantics of SHACL schemas then follows Section 2.4. Importantly, SHACL selectors always select a finite subset of $\mathcal{N} \cup \mathcal{V}$: the selected nodes or values come either from the selector itself, in the case of $\mathrm{is}(c)$, or from $\mathcal{G}$, in the remaining four cases. For example, $\exists p.\top$ selects those nodes of $\mathcal{G}$ that have an outgoing $p$-edge in $\mathcal{G}$—it is grounded to $\mathcal{G}$ in the second line of Table 1. In consequence, each pair $(sel, \varphi)$ in a SHACL schema tests the inclusion of a finite set of nodes or values in a possibly infinite set.

*Example 3.* For better readability we write $\exists\pi$ instead of $\exists^{\geq 1}\pi.\top$ (that is, we omit $\top$) and $\forall\pi.\varphi$ instead of $\exists^{\leq 0}\pi.\neg\varphi$. Let us see how the constraints from Example 2 can be handled in SHACL. For (C1), we assume the value type $\mathtt{int}$ with the obvious meaning. The following SHACL constraints express the constraints (C1–C5):

$$\exists card^- \Rightarrow \mathrm{test}(\mathtt{int}) \tag{C1}$$
$$\exists ownsAccount \Rightarrow \exists email \tag{C2}$$
$$\exists email^- \Rightarrow \exists^{\leq 1} email^- \tag{C3}$$
$$\exists card \Rightarrow (\exists privileged.\neg\mathrm{is}(true)) \vee$$
$$\forall hasAcccess^-.(\exists privileged.\mathrm{is}(true)) \tag{C4}$$

**Table 2: Semantics of a SHACL shape $\varphi$.**

| $\varphi$ | $\mathcal{G}, v \vDash \varphi$ if: |
|---|---|
| $\top$ | trivially satisfied |
| $\neg \varphi$ | not $\mathcal{G}, v \vDash \varphi$ |
| $\varphi \wedge \varphi'$ | $\mathcal{G}, v \vDash \varphi$ and $\mathcal{G}, v \vDash \varphi'$ |
| $\varphi \vee \varphi'$ | $\mathcal{G}, v \vDash \varphi$ or $\mathcal{G}, v \vDash \varphi'$ |
| $\mathrm{is}(c)$ | $v = c$ |
| $\mathrm{test}(\mathbb{v})$ | $v \in [\![\mathbb{v}]\!]$ |
| $\mathrm{closed}(Q)$ | $\forall p \in (\mathcal{P} \cup \mathcal{K}) \setminus Q : \mathrm{not}\ \mathcal{G}, v \vDash \exists^{\geq 1} p.\top$ |
| $\mathrm{eq}(\pi, p)$ | $\{u \mid (v, u) \in [\![\pi]\!]^{\mathcal{G}}\} = \{u \mid (v, u) \in [\![p]\!]^{\mathcal{G}}\}$ |
| $\mathrm{disj}(\pi, p)$ | $\{u \mid (v, u) \in [\![\pi]\!]^{\mathcal{G}}\} \cap \{u \mid (v, u) \in [\![p]\!]^{\mathcal{G}}\} = \emptyset$ |
| $\exists^{\geq n} \pi.\varphi$ | $\#\{u \mid (v, u) \in [\![\pi]\!]^{\mathcal{G}} \wedge \mathcal{G}, u \vDash \varphi\} \geq n$ |
| $\exists^{\leq n} \pi.\varphi$ | $\#\{u \mid (v, u) \in [\![\pi]\!]^{\mathcal{G}} \wedge \mathcal{G}, u \vDash \varphi\} \leq n$ |

$$\exists email \Rightarrow \exists^{\leq 5}\mathrm{hasAcccess}. \tag{C5}$$

Concerning constraint (C3), notice that by using inverse *email* edges, the constraint indeed states that the email addresses uniquely identify users.

The constructs $\mathrm{eq}(\pi, p)$ and $\mathrm{disj}(\pi, p)$ are unique to SHACL. Let us see them in use.

*Example 4.* Using $\mathrm{eq}(\pi, p)$, we can say, for instance, that an owner of an account also has access to it:

$$\exists \mathrm{ownsAccount} \Rightarrow \mathrm{eq}(\mathrm{hasAcccess} \cup \mathrm{ownsAccount}, \mathrm{hasAcccess}).$$

Note how we use eq and $\cup$ to express that the existence of one path (ownsAccount) implies the existence of another path (hasAcccess) with the same endpoints.

A key feature in SHACL that is not available in ShEx is the ability to use regular expressions to talk about complex paths. This provides a limited form of recursive navigation in the graph, even though the standard SHACL does not support recursive constraints (in contrast to standard ShEx). See below for an example.

*Example 5.* Suppose that in Figure 1, we wanted to express that a privileged user may only invite other privileged users, who in turn can also only invite other privileged users. One way to express this in SHACL is as follows:

$$\exists privileged \Rightarrow \exists privileged.\mathrm{is}(false)\vee$$
$$\forall(\mathrm{invited}^* \cdot privileged).\mathrm{is}(true).$$

## 4 SHEX ON COMMON GRAPHS

While SHACL is conceptually the simplest of the three language, ShEx lies at the opposite end of the spectrum. It is an intricate, mutually recursive combination of a simple logic for shapes and a powerful formalism (triple expressions) for generating the allowed neighbourhoods. In this work we consider non-recursive ShEx, where shapes and triple expressions can be nested multiple times, but cannot be actually recursive. We choose non-recursive ShEx because it is significantly easier to understand (so it aligns with our overall understandability goal). The abstraction of ShEx over

common graphs is based on the treatment of ShEx on RDF triples by Boneva et al. [8]. The correspondence to standard ShEx is discussed in Appendix C.

DEFINITION 6 (SHAPES AND TRIPLE EXPRESSIONS). *ShEx shapes $\varphi$, triple expressions $e$, and closed triple expressions $f$ are defined by the following grammar*

$$\varphi ::= \mathrm{is}(c) \mid \mathrm{test}(\mathbb{v}) \mid \{e\} \mid \varphi \wedge \varphi \mid \varphi \vee \varphi \mid \neg\varphi.$$
$$e ::= f\,;op_- \quad \mid \quad f\,;op_\pm$$
$$op_- ::= (\neg P^-)^*$$
$$op_\pm ::= (\neg P^-)^*\,;(\neg Q)^*$$
$$f ::= \varepsilon \quad \mid \quad p.\varphi \quad \mid \quad p^-.\varphi \quad \mid \quad f\,;f \quad \mid \quad f\,|\,f \quad \mid \quad f^*.$$

*where $c \in \mathcal{V}$, $\mathbb{v} \in \mathcal{T}$, $p \in \mathcal{P} \cup \mathcal{K}$, and $P, Q \subseteq_{fin} \mathcal{P} \cup \mathcal{K}$.*

The notion of satisfaction for ShEx shapes and the semantics of triple expressions are defined by mutual recursion in Table 3 and Table 4. Triple expressions are used to specify neighbourhoods of nodes and values. They require to consider incoming and outgoing edges separately. For this purpose we decorate incoming edges with $^-$. Formally, we introduce a fresh predicate $p^-$ for each $p \in \mathcal{P}$ and a fresh key $k^-$ for each $k \in \mathcal{K}$. We let $\mathcal{P}^- = \{p^- \mid p \in \mathcal{P}\}$, $\mathcal{K}^- = \{k^- \mid k \in \mathcal{K}\}$, $\mathcal{E}^- = \mathcal{N} \times \mathcal{P}^- \times \mathcal{N} \cup \mathcal{V} \times \mathcal{K}^- \times \mathcal{N}$, and define $\mathrm{Neigh}^\pm_{\mathcal{G}}(v) \subseteq \mathcal{E} \cup \mathcal{E}^-$ as

$$\left\{(v, p, v') \mid (v, p, v') \in \mathcal{G}\right\} \cup \left\{(v, p^-, v') \mid (v', p, v) \in \mathcal{G}\right\}.$$

Compared to $\mathrm{Neigh}_{\mathcal{G}}(v)$, apart from flipping the incoming edges and marking them with $^-$, we also represent each loop $(v, p, v)$ twice: once as an outgoing edge $(v, p, v)$ and once as an incoming edge $(v, p^-, v)$. Notice that in Table 4, $\neg P$ and its inverse counterpart $\neg P^-$ are treated as any other triple expressions, even though in the grammar they are allowed only at the top level.

Closed triple expressions $f$ define neighbourhoods that use only a finite number of predicates and keys; such neighbourhoods are also called *closed* in ShEx terminology. General triple expressions $e$ *open* the neighbourhood either only w.r.t. incoming triples ($op_-$) by allowing any incoming triples whose predicate or key is not in a set $P$, or w.r.t. both incoming and outgoing triples ($op_\pm$) by additionally allowing outgoing triples whose predicate or key is not in a set $Q$. Let $\top = \varepsilon\,;(\neg\emptyset^-)^*\,;(\neg\emptyset)^*$. Then $\top$ describes all possible neighbourhoods and $\{\top\}$ is satisfied in every node and in every value of every graph.

DEFINITION 7 (SHEX SELECTORS). *A ShEx selector is a ShEx shape of a restricted form, defined by the grammar*

$$sel ::= \mathrm{is}(c) \mid \{p.\mathrm{is}(c)\,;\top\} \mid \{p^-.\mathrm{is}(c)\,;\top\} \mid$$
$$\{p.\{\top\}\,;\top\} \mid \{p^-.\{\top\}\,;\top\}.$$

*where $p \in \mathcal{P} \cup \mathcal{K}$ and $c \in \mathcal{V}$.*

Following Section 2.4, a *ShEx schema $\mathcal{S}$* is a set of pairs of the form $(sel, \varphi)$ where $\varphi$ is a ShEx shape and $sel$ is a ShEx selector.

We will be using these syntactic short-hands:
- $e^n$ for some positive integer $n$ denotes the triple expression $e\,;\ldots\,;e$ where $e$ is used $n$ times,
- $e^{\leq n}$ as a short-hand for $\varepsilon \mid e^1 \mid \ldots \mid e^n$ and $e^{\geq n}$ for $e^n\,;e^*$,
- for a closed triple expression $f$, we let

$$\{f\}^\circ = \{f\,;(\neg P^-)^*\,;(\neg P)^*\}$$

**Table 3: Satisfaction of ShEx shapes.**

| $\varphi$ | $\mathcal{G}, v \vDash \varphi$ for $v \in \mathcal{N} \cup \mathcal{V}$ |
|---|---|
| $test(\mathbb{v})$ | $v \in [\![\mathbb{v}]\!]$ |
| $is(c)$ | $v = c$ |
| $\{e\}$ | $\mathrm{Neigh}_{\mathcal{G}}^{\pm}(v) \in [\![e]\!]_v^{\mathcal{G}}$ |
| $\varphi_1 \wedge \varphi_2$ | $\mathcal{G}, v \vDash \varphi_1$ and $\mathcal{G}, v \vDash \varphi_2$ |
| $\varphi_1 \vee \varphi_2$ | $\mathcal{G}, v \vDash \varphi_1$ or $\mathcal{G}, v \vDash \varphi_2$ |
| $\neg\varphi$ | not $\mathcal{G}, v \vDash \varphi$ |

**Table 4: Semantics of triple expressions.**

| $e$ | $[\![e]\!]_v^{\mathcal{G}} \subseteq \mathcal{E} \cup \mathcal{E}^-$ |
|---|---|
| $\varepsilon$ | $\{\emptyset\}$ |
| $p.\varphi$ | $\left\{ \{(v, p, v')\} \subseteq \mathcal{E} \mid \mathcal{G}, v' \vDash \varphi \right\}$ |
| $p^-.\varphi$ | $\left\{ \{(v, p^-, v')\} \subseteq \mathcal{E}^- \mid \mathcal{G}, v' \vDash \varphi \right\}$ |
| $\neg P$ | $\left\{ \{(v, p, v')\} \subseteq \mathcal{E} \mid p \notin P \right\}$ |
| $\neg P^-$ | $\left\{ \{(v, p^-, v')\} \subseteq \mathcal{E}^- \mid p \notin P \right\}$ |
| $e_1 ; e_2$ | $\left\{ T_1 \cup T_2 \mid T_1 \in [\![e_1]\!]_v^{\mathcal{G}}, T_2 \in [\![e_2]\!]_v^{\mathcal{G}}, T_1 \cap T_2 = \emptyset \right\}$ |
| $e_1 \mid e_2$ | $[\![e_1]\!]_v^{\mathcal{G}} \cup [\![e_2]\!]_v^{\mathcal{G}}$ |
| $e^*$ | $\{\emptyset\} \cup \bigcup_{n=1}^{\infty} \left\{ T_1 \cup \cdots \cup T_n \mid \begin{array}{l} T_1, \ldots, T_n \in [\![e]\!]_v^{\mathcal{G}} \text{ and} \\ T_i \cap T_j = \emptyset \text{ for all } i \neq j \end{array} \right\}$ |

where $P$ is the set of predicates and keys that appear *directly* in $f$ without considering those that appear in $\varphi$ for a sub-expression of the form $p.\varphi$. For instance, if $f = p.\{p'.is(c)\}$, then $P = \{p\}$. Also, $P^-$ is the set of predicates and keys that appear inversed in $f$.

*Example 6.* Let us now see how the concrete constraints from Example 2 can be handled in ShEx.

$$\{card^-.\{\top\};\top\} \Rightarrow test(\mathbb{int}) \tag{C1}$$

$$\{ownsAccount.\{\top\};\top\} \Rightarrow \{email.\{\top\}\}^\circ \tag{C2}$$

$$\{email^-.\{\top\};\top\} \Rightarrow \left\{(email^-.\{\top\})^{\leq 1}\right\}^\circ \tag{C3}$$

$$\{card.\{\top\};\top\} \Rightarrow \{privileged.\neg is(true)\}^\circ \vee$$
$$\left\{(hasAcccess^-.\{privileged.is(true)\}^\circ)^*\right\}^\circ \tag{C4}$$

$$\{email.\{\top\};\top\} \Rightarrow \left\{(hasAcccess.\{\top\})^{\leq 5}\right\}^\circ \tag{C5}$$

We next show a more complex example, which illustrates the power of ShEx that is not readily available in SHACL or PG-Schema.

*Example 7.* Suppose that we want to express the following constraint on each user who owns an account: the number of accounts to which the user has access is greater or equal to the number of accounts that the user owns. We can do this in ShEx as follows:

$$\{ownsAccount.\{\top\};\top\} \Rightarrow$$

$$\left\{(hasAcccess.\{\top\})^*;(ownsAccount.\{\top\};hasAcccess.\{\top\})^*\right\}^\circ$$

Finally, let us see why ShEx and SHACL count differently.

*Example 8 (ShEx counts edges).* The following SHACL schema expresses that from every node with an outgoing hasAcccess-edge,

**Table 5: Semantics of content types.**

| $\mathbb{c}$ | $[\![\mathbb{c}]\!] \subseteq \mathcal{R}$ |
|---|---|
| $[\![\top]\!]$ | $\mathcal{R}$ |
| $[\![\{\}]\!]$ | $\{\mathbf{r}_\emptyset\}$ |
| $[\![\{k : \mathbb{v}\}]\!]$ | $\left\{ \{(k, w)\} \mid w \in [\![\mathbb{v}]\!] \right\}$ |
| $[\![\mathbb{c}_1 \,\&\, \mathbb{c}_2]\!]$ | $\left\{ (r_1 \cup r_2) \in \mathcal{R} \mid r_1 \in [\![\mathbb{c}_1]\!] \wedge r_2 \in [\![\mathbb{c}_2]\!] \right\}$ |
| $[\![\mathbb{c}_1 \mid \mathbb{c}_2]\!]$ | $[\![\mathbb{c}_1]\!] \cup [\![\mathbb{c}_2]\!]$ |

there should be exactly two nodes accessible via a hasAcccess-edge or an ownsAccount-edge:

$$\exists hasAcccess \Rightarrow \exists^{=2}(hasAcccess \cup ownsAccount).\top;$$

here $\exists^{=n}\pi.\varphi$ is a shorthand for $\exists^{\leq n}\pi.\varphi \wedge \exists^{\geq n}\pi.\varphi$. For instance, the graph below on the left is valid, whereas the one on the right is not.

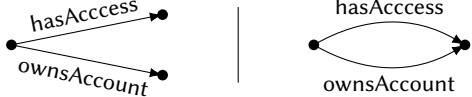

The same constraint cannot be expressed in ShEx because ShEx cannot distinguish these two graphs. Indeed, ShEx triple expressions count triples adjacent to a node, whereas SHACL and PG-Schema count nodes on the opposite end of such triples.

# 5 PG-SCHEMA ON COMMON GRAPHS

PG-Schema is a non-recursive combination of a logic and two generative formalisms. It uses path expressions to specify paths (as in SHACL), and *content types* to specify node contents. Both path expressions and content types are then used in formulas defining shapes. Content types in PG-Schema play a role similar to triple expressions in ShEx, but they are only used for properties. Because all properties of a node must have different keys, they are much simpler than triple expressions (in fact, they can be translated into a fragment of SHACL). Unlike for SHACL and ShEx, the abstraction of PG-Schema on common graphs departs significantly from the original design. Original PG-Schema uses queries written in an external query language, which is left unspecified aside from some basic assumptions about the expressive power. Here we use a specific query language (PG-path expressions). Importantly, up to the choice of the query language, the abstraction we present here faithfully captures the expressive power of the original PG-Schema. A detailed comparison can be found in Appendix D.

**DEFINITION 8 (CONTENT TYPE).** *A content type is an expression $\mathbb{c}$ of the form defined by the grammar*

$$\mathbb{c} ::= \top \ \Big| \ \{\} \ \Big| \ \{k : \mathbb{v}\} \ \Big| \ \mathbb{c} \,\&\, \mathbb{c} \ \Big| \ \mathbb{c} \mid \mathbb{c}.$$

*where $k \in \mathcal{K}$ and $\mathbb{v} \in \mathcal{T}$.*

Recall that $\mathcal{R}$ is the set of all records (finite-domain partial functions $r : \mathcal{K} \rightharpoonup \mathcal{V}$). We write $\mathbf{r}_\emptyset$ for the empty record. For records $r_1$ and $r_2$, we let $r_1 \cup r_2$ be the function that behaves as $r_1$ on $\mathrm{dom}(r_1)$ and as $r_2$ on $\mathrm{dom}(r_2)$. We require that $r_1(k) = r_2(k)$ for every $k \in \mathrm{dom}(r_1) \cap \mathrm{dom}(r_2)$. The semantics of content types is defined in Table 5. Note that $[\![\mathbb{c}]\!]$ is independent from $\mathcal{G}$ and can be infinite.

**Table 6: Semantics of PG-path expressions.**

| $\pi$ | $[\![\pi]\!]^{\mathcal{G}} \subseteq (\mathcal{N} \cup \mathcal{V}) \times (\mathcal{N} \cup \mathcal{V})$ for $\mathcal{G} = (E, \rho)$ |
|---|---|
| $\{k : c\}$ | $\{(u, u) \mid u \in \text{Nodes}(\mathcal{G}) \wedge (k, c) \in \rho(u)\}$ |
| $\neg\{k : c\}$ | $\{(u, u) \mid u \in \text{Nodes}(\mathcal{G}) \wedge (k, c) \notin \rho(u)\}$ |
| $\mathbb{c}$ | $\{(u, u) \mid u \in \text{Nodes}(\mathcal{G}) \wedge \rho(u) \in [\![\mathbb{c}]\!]\}$ |
| $\neg\mathbb{c}$ | $\{(u, u) \mid u \in \text{Nodes}(\mathcal{G}) \wedge \rho(u) \notin [\![\mathbb{c}]\!]\}$ |
| $k$ | $\{(u, w) \mid \rho(u, k) = w\}$ |
| $p$ | $\{(u, v) \mid (u, p, v) \in E\}$ |
| $\neg P$ | $\{(u, v) \mid \exists p : (u, p, v) \in E \wedge p \notin P\}$ |
| $\pi^-$ | $\{(u, v) \mid (v, u) \in [\![\pi]\!]^{\mathcal{G}}\}$ |
| $\pi \cdot \pi'$ | $\{(u, v) \mid \exists w : (u, w) \in [\![\pi]\!]^{\mathcal{G}} \wedge (w, v) \in [\![\pi']\!]^{\mathcal{G}}\}$ |
| $\pi \cup \pi'$ | $[\![\pi]\!]^{\mathcal{G}} \cup [\![\pi']\!]^{\mathcal{G}}$ |
| $\pi^*$ | $\{(u, u) \mid u \in \text{Nodes}(\mathcal{G})\} \cup [\![\pi]\!]^{\mathcal{G}} \cup [\![\pi \cdot \pi]\!]^{\mathcal{G}} \cup \dots$ |

*Example 9.* We assume integers and strings are represented via $\text{int}, \text{str} \in \mathcal{T}$. Suppose we want to create a content type for nodes that have a string value for the *email* key and *optionally* have an integer value for the *card* key. No other key-value pairs are allowed. We should then use $\{email : \text{str}\} \& (\{card : \text{int}\} \mid \{\})$.

**Definition 9 (PG-path expressions).** *A PG-path expression is an expression $\pi$ of the form defined by the grammar*

$$\pi ::= \bar{\pi} \mid \bar{\pi} \cdot k \mid k^- \cdot \bar{\pi} \mid k^- \cdot \bar{\pi} \cdot k' \,.$$

$$\bar{\pi} ::= \{k : c\} \mid \neg\{k : c\} \mid \mathbb{c} \mid \neg\mathbb{c} \mid p \mid \neg P \mid \bar{\pi}^- \mid \bar{\pi} \cdot \bar{\pi} \mid \bar{\pi} \cup \bar{\pi} \mid \bar{\pi}^* \,.$$

*where $k, k' \in \mathcal{K}$, $c \in \mathcal{V}$, $\mathbb{c}$ is a content type, $p \in \mathcal{P}$, and $P \subseteq_{fin} \mathcal{P}$. We use $k$, $k^-$, and $k^- \cdot k'$ as short-hands for PG-path expressions $\top \cdot k$, $k^- \cdot \top$, and $k^- \cdot \top \cdot k'$, respectively.*

Unlike in SHACL, PG-path expressions cannot navigate freely through values. In the property graph world, this would correspond to a join, which is a costly operation. Indeed, existing query languages for property graphs do not allow joins under $^*$. However, PG-path expressions can start in a value and finish in a value. This leads to *node-to-node*, *node-to-value*, *value-to-node*, and *value-to-value* PG-path expressions, reflected in the four cases in the first rule of the grammar.

The semantics of PG-path expression $\pi$ for graph $\mathcal{G}$ is a binary relation over $\text{Nodes}(\mathcal{G}) \cup \text{Values}(\mathcal{G})$, defined in Table 6. In the table, $k$ is treated as any other subexpressions, eventhough it can only be used at the end of a PG-path expression, or in the beginning as $k^-$. Notice that $\neg\mathbb{c}$ matches nodes whose content is not of type $\mathbb{c}$, $\neg P$ matches edges with a label that is not in $P$ (in particular, $\neg\emptyset$ matches all edges). Also, $[\![\pi]\!]^{\mathcal{G}}$ is always a subset of $\mathcal{N} \times \mathcal{N}$, $\mathcal{N} \times \mathcal{V}$, $\mathcal{V} \times \mathcal{N}$, or $\mathcal{V} \times \mathcal{V}$, corresponding to the four kinds of PG-path expressions discussed above.

**Definition 10 (PG-Shapes).** *A PG-Shape is an expression $\varphi$ defined by the following grammar:*

$$\varphi ::= \exists^{\leq n} \pi \mid \exists^{\geq n} \pi \mid \varphi \wedge \varphi \,.$$

*where $\pi$ is a PG-path expression. We use $\exists$ and $\nexists$ as short-hands for $\exists^{\geq 1}$ and $\exists^{\leq 0}$.*

**Table 7: Satisfaction of PG-shapes**

| $\varphi$ | $\mathcal{G}, v \vDash \varphi$ for $v \in \mathcal{N} \cup \mathcal{V}$ |
|---|---|
| $\exists^{\leq n} \pi$ | $\#\{v' \mid (v, v') \in [\![\pi]\!]^{\mathcal{G}}\} \leq n$ |
| $\exists^{\geq n} \pi$ | $\#\{v' \mid (v, v') \in [\![\pi]\!]^{\mathcal{G}}\} \geq n$ |
| $\varphi_1 \wedge \varphi_2$ | $\mathcal{G}, v \vDash \varphi_1$ and $\mathcal{G}, v \vDash \varphi_2$ |

The semantics of PG-shapes is defined in Table 7. We say $v \in \mathcal{N} \cup \mathcal{V}$ *satisfies* a PG-shape $\varphi$ in a graph $\mathcal{G}$ if $\mathcal{G}, v \vDash \varphi$. Every PG-shape is satisfied by nodes only or by values only.

**Definition 11 (PG-Selectors).** *A PG-selector is a PG-shape of the form $\exists \pi$.*

A *PG-Schema* $\mathcal{S}$ is a finite set of pairs $(sel, \varphi)$ where $sel$ is a PG-selector and $\varphi$ is a PG-shape. The semantics of PG-Schemas is defined just like in Section 2.4.

*Example 10.* The constraints (C1-C5) from Example 2 can be handled in PG-Schema as follows:

$$\exists card^- \Rightarrow \exists(\{card : \text{int}\} \& \top) \tag{C1}$$

$$\exists ownsAccount \Rightarrow \exists email \tag{C2}$$

$$\exists email^- \Rightarrow \exists^{\leq 1} email^- \tag{C3}$$

$$\exists(\{card : \text{any}\} \& \top) \cdot \{privileged : true\} \Rightarrow$$
$$\nexists hasAcccess^- \cdot \neg\{privileged : true\} \tag{C4}$$

$$\exists email \Rightarrow \exists^{\leq 5} hasAcccess \tag{C5}$$

A characteristic feature of PG-Schema, revealing its database provenience, is that it can close the whole graph by imposing restrictions on all nodes.

*Example 11.* Given a common graph such as the one in Figure 1, we might want to express that each node has a key *privileged* with a boolean value and either a key *card* with an integer value or a key *email* with a string value, and no other keys are allowed. In PG-Schema this can be expressed as follows:

$$\exists\top \Rightarrow \exists\{privileged : \text{bool}\} \& (\{card : \text{int}\} \mid \{email : \text{str}\}) \,.$$

We can also forbid any predicates except those mentioned in the running example:

$$\exists\top \Rightarrow \nexists\neg\{ownsAccount, hasAcccess, invited\} \,.$$

# 6 COMMON GRAPH SCHEMA LANGUAGE

We now present the Common Graph Schema Language (CoGSL), which combines the core functionalities shared by SHACL, ShEx, and PG-Schema (over common graphs).

Let us begin by examining the restrictions that need to be imposed. We shall refer to shapes and selectors used in CoGSL as *common shapes* and *common selectors*.

Common shapes cannot be closed under disjunction and negation, because PG-Schema shapes are purely conjunctive. For the same reason common shapes cannot be nested.

Kleene star $^*$ cannot be allowed in path expressions because we consider ShEx without recursion. By switching to ShEx with recursion, we would be able to support arbitrary SHACL path expressions

in shapes of the form $\exists \pi$, but not arbitrary PG-path expressions as these are too expressive for SHACL.

Supporting path expressions traversing more than one edge under counting quantifiers is impossible as this would not be expressible in ShEx. Supporting disjunctions of labels of the form $p_1 \cup p_2$ is also impossible, due to a mismatch in the approach to counting: while SHACL and PG-Schema count nodes and values, ShEx counts triples, as illustrated in Example 8.

Closed content types and $\neg P$ cannot be used freely, because neither SHACL nor ShEx are capable of closing only properties or only predicate edges: both must be closed at the same time.

Finally, selectors are restricted because SHACL and ShEx do not support $\top$ as a selector; that is, one cannot say that each node (or value) in the graph satisfies a given shape. This means that SHACL and ShEx schemas always allow a disconnected part of the graph that uses only predicates and keys not mentioned in the schema, whereas PG-Schema can disallow it (see Example 11).

Putting these restrictions together we obtain the Common Graph Schema Language. We define it below as a fragment of PG-Schema.

**Definition 12 (common shape).** *A common shape $\varphi$ is an expression given by the grammar*

$$\varphi ::= \exists \pi \mid \exists^{\leq n} \pi_1 \mid \exists^{\geq n} \pi_1 \mid \exists \mathbb{c} \wedge \not\exists \neg P \mid \varphi \wedge \varphi \ .$$

$$\mathbb{c} ::= \{\} \ \mid \ \{k : \mathbb{v}\} \ \mid \ \mathbb{c} \ \& \ \mathbb{c} \ \mid \ \mathbb{c} \mid \mathbb{c} \ .$$

$$\pi_0 ::= \{k : c\} \ \mid \ \neg\{k : c\} \ \mid \ \mathbb{c} \ \& \ \top \ \mid \ \neg(\mathbb{c} \ \& \ \top) \ \mid \ \pi_0 \cdot \pi_0 \ .$$

$$\pi_1 ::= \pi_0 \cdot p \cdot \pi_0 \ \mid \ \pi_0 \cdot p^- \cdot \pi_0 \ \mid \ \pi_0 \cdot k \ \mid \ k^- \cdot \pi_0 \ .$$

$$\bar{\pi} ::= \pi_0 \ \mid \ p \ \mid \ \bar{\pi}^- \ \mid \ \bar{\pi} \cdot \bar{\pi} \ \mid \ \bar{\pi} \cup \bar{\pi} \ .$$

$$\pi ::= \bar{\pi} \ \mid \ \bar{\pi} \cdot k \ \mid \ k^- \cdot \bar{\pi} \ \mid \ k^- \cdot \bar{\pi} \cdot k' \ .$$

*where $n \in \mathbb{N}$, $P \subseteq_{fin} \mathcal{P}$, $k, k' \in \mathcal{K}$, $c \in \mathcal{V}$, and $p \in \mathcal{P}$.*

That is, $\mathbb{c}$ is a content type that does not use $\top$ (a *closed* content type), $\pi_0$ is a PG-path expression that always stays in the same node (a *filter*), $\pi_1$ is a PG-path expression that traverses a single edge or property (forward or backwards), and $\pi$ is a PG-path expression that uses neither $^*$ nor $\neg P$. Moreover, $\pi_0$, $\pi_1$, and $\pi$ can only use *open* content types; that is, content types of the form $\mathbb{c} \ \& \ \top$. The use of $\neg P$ is limited to closing the neighbourhood of a node (this is the only way PG-Schema can do it).

**Definition 13 (common selector).** *A common selector is a common shape of one of the following forms*

$$\exists k, \ \exists p \cdot \pi, \ \exists p^- \cdot \pi, \ \exists \{k : c\} \cdot \pi, \ \exists (\{k : \mathbb{v}\} \ \& \ \top) \cdot \pi, \ \exists k^- \cdot \pi,$$

*where $k \in \mathcal{K}$, $p \in \mathcal{P}$, $c \in \mathcal{V}$, $\mathbb{v} \in \mathcal{T}$ and $\pi = \bar{\pi}$ or $\pi = \bar{\pi} \cdot k'$ for some PG-path expression $\bar{\pi}$ generated by the grammar in Definition 12 and some $k' \in \mathcal{K}$.*

That is, a common selector is a common shape of the form $\exists \pi$ such that the PG-path expression $\pi$ requires the focus node or value to occur in a triple with a specified predicate or key.

A *common schema* is a finite set of pairs $(sel, \varphi)$ where $sel$ is a common selector and $\varphi$ is a common shape. The semantics is inherited from PG-Schema.

We note that we showed that the constraints (C1)-(C5) from our running example can be expressed in all three formalisms. Specifically, the PG-Schema representation from Example 10 is also a common schema.

**Proposition 1.** *For every common schema there exist equivalent SHACL and ShEx schemas.*

The translation is relatively straightforward (see Appendix E). The two main observations are that star-free PG-path expressions can be simulated by nested SHACL and ShEx shapes, and that closure of SHACL and ShEx shapes under Boolean connectives allows encoding complex selectors in the shape (as the antecedent of an implication). We illustrate the latter in Example 12.

*Example 12 (Complex paths in selectors).* We want to express that all users who have invited a user who has invited someone (so there is a path following two invited edges) must have a key *email* of type $\mathtt{str}$. In PG-schema we express this as:

$$\exists \text{invited} \cdot \text{invited} \Rightarrow \{email : \mathtt{str}\} \ \& \ \top$$

At first glance, it seems unclear how to express this in the other formalisms, since they do not permit paths in the selector. However, we can see that paths in selectors can be encoded into the shape: In SHACL, using the same example, we do this by

$$\exists \text{invited} \Rightarrow \neg(\exists \text{invited} \cdot \text{invited}) \vee \exists email.\text{test}(\mathtt{str})$$

And in ShEx for this example would be:

$$\{\text{invited}.\{\top\} ; \top\} \Rightarrow \neg\varphi_2 \vee \{email.\text{test}(\mathtt{str})\}^\circ$$

where $\varphi_2 = \left\{ (\text{invited}.\varphi_1)^{\geq 1} \right\}^\circ$ and $\varphi_1 = \left\{ \text{invited}.\{\top\}^{\geq 1} \right\}^\circ$. That is, $\varphi_1$ is satisfied by nodes that have an outgoing path invited, and $\varphi_2$ by nodes that have an outgoing path invited $\cdot$ invited. For paths of unbounded length, it is not apparent how such a translation would proceed for ShEx schemas in the absence of recursion.

## 7 RELATED WORK

*SHACL literature.* The authoritative source of SHACL is the W3C recommendation [29]. Further literature on SHACL following its standardisation can be roughly divided into two groups. The first studies the formal properties and expressiveness of the non-recursive fragment [7]. Notable examples in this category (in no particular order) is the work from Delva et al. on data provenance [16], by Pareti et al. on satisfiability and (shape) containment [40] and the work of Leinberger et al. connecting the containment problem to description logics [34]. The other body of work on SHACL is concerned with either proposing a suitable semantics for the recursive fragment [1, 5, 12, 13] or assuming a given one and studying the complexity of certain problems in their chosen recursive setting [39]. First reports on practical applications and use cases for SHACL include, for instance, the expressivity of property constraints, or mining and extracting constraints in the context of practical large KGs such as Wikidata and DBpedia [18, 44].

*ShEx literature.* ShEx was initially proposed in 2014 as a concise and human-readable language to describe, validate, and transform RDF data [43]. Its formal semantics was formally defined in [48]. The semantics of ShEx schemas combining recursion and negation was later presented in [8]. The current semantic specification of the ShEx language has been published as a W3C Community group report [42] and a new language version is currently being defined as part of the IEEE Working group on Shape Expressions[1]. As for practical applications, ShEx has been applied as a descriptive schema

---

[1]https://shex.io/shex-next/

language through the Wikidata Schemas project[2]. Additional work went into extending ShEx to handle graph models that go beyond RDF, like WShEx to validate Wikibase graphs [31], ShEx-Star to handle RDF-Star and PShEx to handle property graphs [30].

*PG-Schema literature.* PG-Schema, as introduced in [2], builds on PG-Types and PG-Keys [3] to enhance schema support for property graphs. Despite limited schema support in current systems and the first GQL standard [25], PG-Schema combines flexible type definitions via PG-Types with expressive key constraints from PG-Keys. This formalism provides a robust syntax and semantics for property graph management, aiming to inspire future versions of the GQL standard and broaden the capabilities of graph database systems.

*Works that compare RDF schema formalisms.* In Chapter 7 of [20], the authors compare common features and differences between ShEx and SHACL and [32] presents a simplified language called S, which captures the essence of ShEx and SHACL. Tomaszuk [49] analyzes advancements in RDF validation, highlighting key requirements for validation languages and comparing the strengths and weaknesses of various approaches.

*Interoperability between schema graph formalisms.* Interoperability between schema graph formalisms like RDF and Property Graphs remains challenging due to differences in structure and semantics. RDF focuses on triple-based modeling with formal semantics, while Property Graphs allow flexible annotation of relationships with properties. RDF-star [22] and RDF 1.2 [27] extend RDF 1.1 by enabling statements about triples, aligning more closely with LPG: for instance, RDF-star allows triples to function as subjects or objects, similar to how LPG edges carry properties.

It should be noted here that by adopting *named graphs* [11], already RDF 1.1 provided a mechanism for making statements about (sub-)graphs; likewise, different *reification* mechanisms have been proposed in the literature for RDF in order to "embed" meta-statements about triples (and graphs) in "vanilla" RDF graphs, ranging from the relatively verbose original W3C reification vocabulary as part of the original RDF specification, to more subtle approaches such as singleton property reification [36], which is pretty close to the unique identifiers used for edges in most LPG models. Lastly, custom reification models are used, for instance, in Wikidata, to map Wikibase's property graph schema to RDF, cf. e.g. [18, 23]. All these approaches, in principle, facilitate general or specific mappings between RDF and LPGs.

Contrary to such mappings using named graphs or reification, there have been several prior proposals for uniying graph data models. The OneGraph initiative [33] also aims to bridge the different graph data models, by promoting a unified graph data model for seamless interaction. Similarly, MilleniumDB's Domain Graph model [50] aims at providing a general graph model bridging between RDF, RDF-star and property graphs. Work on mappings [4] has also explored schema-independent and schema-dependent methods for transforming RDF into Property Graphs, providing formal foundations for preserving information and semantics.

*Schemas for tree-structured data.* The principle of defining (parts of) schemas as a set of pairs $(sel, \varphi)$ is also used in schema languages for XML. A DTD [10] is essentially such a set of pairs in which $sel$ selects nodes with a certain label, and $\varphi$ describes the structure of their children. In XML Schema, the principle was used for defining key constraints (using *selectors* and *fields*) [19, Section 3.11.1]. The equally expressive language BonXai [35] is based on writing the entire schema using such rules. Schematron [24] is another XML schema language that differs from grammar-based languages by defining patterns of assertions using XPath expressions [17]. It excels in specifying constraints across different branches of a document tree, where traditional schema paradigms often fall short. Schematron's rule-based structure, composed of phases, patterns, rules, and assertions, allows for the validation of documents.

*RDF validation.* Last, but not least, it should be noted that the requirement for (constraining) schema languages—besides ontology languages such as OWL and RDF Schema—in the Semantic Web community is much older than the more recent additions of SHACL and ShEx. Earlier proposals in a similar direction include efforts to add constraint readings of Description Logic axioms to OWL, such as OWL Flight [15] or OWL IC [47]. Another approach is Resource Shapes (ReSh) [45], a vocabulary for specifying RDF shapes. The authors of ReSh recognize that RDF terms originate from various vocabularies, and the ReSh shape defines the integrity constraints that RDF graphs are required to satisfy. Similarly, Description Set Profiles (DSP) [37] and SPARQL Inferencing Notation (SPIN) [28] are notable alternatives. While SHACL, ShEx, and ReSh share declarative, high-level descriptions of RDF graph content, DSP and SPIN offer additional mechanisms for validating and constraining RDF data, each with its own strengths and applications.

## 8 CONCLUSIONS

We provided a formal and comprehensive comparison of the three most prominent schema languages in the Semantic Web and Graph Database communities: SHACL, ShEx, and PG-Schema. Through painstaking discussions within our working group, we managed to (1) agree on a common data model that captures features of both RDF and Property Graphs and (2) extract, for each of the languages, a core that we mutually agree on, which we define formally. Moreover, the definitions of (the cores of) each of the schema languages on a common formal framework allows readers to maximally leverage their understanding of one schema language in order to understand the others. Furthermore, this common framework allowed us to extract the Common Graph Schema Language, which is a cleanly defined set of functionalities shared by SHACL, ShEx, and PG-Schema. This commonality can serve as a basis for future efforts in integrating or translating between the languages, promoting interoperability in applications that rely on heterogeneous data models. For example, we want to investigate recursive ShEx and more expressive query languages for PG-Schema more deeply.

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

## A  DISTILLING THE COMMON DATA MODEL

In this section we discuss the relationship between common graphs and the standard data models of the three schema formalisms formalisms—RDF and property graphs.

### A.1  Comparison with RDF

As explained in Section 2, common graphs can be naturally seen as finite sets of triples from $\mathcal{E} = (\mathcal{N} \times \mathcal{P} \times \mathcal{N}) \cup (\mathcal{N} \times \mathcal{K} \times \mathcal{V})$, with $(E, \rho)$ corresponding to $E \cup \{(u, k, v) \mid \rho(u, k) = v\}$.

Unlike in RDF, a common graph may contain at most one tuple of the form $(u, k, v)$ for each $u \in \mathcal{N}$ and $k \in \mathcal{K}$. This reflects the assumption that properties are single-valued, which is present in the property graph data model.

In the RDF context, one would assume the following:

- $\mathcal{N} \subseteq$ IRIs $\cup$ Blanks,
- $\mathcal{P} \subseteq$ IRIs,
- $\mathcal{K} \subseteq$ IRIs,
- $\mathcal{V} =$ Literals.

However, the common graph data model does not refer to IRIs, Blanks, and Literals at all, because these are not part of the property graph data model.

In contrast to the RDF model, but in accordance with the perspective commonly taken in databases, both values and nodes are atomic. For nodes we completely abstract away from the actual representation of their identities. We do not even distinguish between IRIs and Blanks. An immediate consequence of this is that schemas do not have access to any information about the node other than the triples in which it participates. In particular, they cannot compare nodes with constants. This is a significant restriction with respect to the RDF data model, but it follows immediately from the same assumption made in the property graph data model. On the positive side, this aspect is entirely orthogonal to the main discussion in this paper, so eliminating it from the common data model does not oversimplify the picture.

For values we take a more subtle approach: we assume a set $\mathcal{T}$ of value types, with each $\mathbb{v} \in \mathcal{T}$ representing a set $[\![\mathbb{v}]\!] \subseteq \mathcal{V}$. This captures uniformly data types, such as integer or string, and user-defined checks, such as interval bounds for numeric values or regular expressions for strings. On the other hand, the common graph data model does not include any binary relations over values, such as an order.

### A.2  Comparison with property graphs

Let us recall the standard definition of property graphs [2].

DEFINITION 14 (PROPERTY GRAPH). *A property graph is a tuple* $(N, E, \pi, \lambda, \rho)$ *such that*

- *N is a finite set of nodes;*
- *E is a finite set of edges, disjoint from N;*
- *$\pi : E \to (N \times N)$ maps edges to their source and target;*
- *$\lambda : (N \cup E) \to 2^{\mathcal{P}}$ maps nodes and edges to finite sets of labels;*
- *$\rho : (N \cup E) \times \mathcal{K} \rightharpoonup \mathcal{V}$ is a finite-domain partial function mapping element-key pairs to values.*

A common graph $G = (E', \rho')$ can be easily represented as a property graph by letting

- $N = \text{Nodes}(G)$,
- $E = E'$,
- $\pi = \{(e, (v_1, v_2)) \mid e = (v_1, p, v_2) \in E\}$,
- $\lambda = \{(e, \{p\}) \mid e = (v_1, p, v_2) \in E\} \cup \{(v, \emptyset) \mid v \in N\}$, and
- $\rho = \rho'$.

It is possible to characterise exactly the property graphs that are such representations of common graphs. These are the property graphs $(N, E, \pi, \lambda, \rho)$ for which it holds that:

(1) $\lambda(v) = \emptyset$ for all $v \in N$, and $\lambda(e)$ is a singleton for all $e \in E$,
(2) there cannot be two distinct edges $e_1, e_2 \in E$ such that $\pi(e_1) = \pi(e_2)$ and $\lambda(e_1) = \lambda(e_2)$, and
(3) $\rho(e, k)$ is undefined for all $e \in E, k \in \mathcal{K}$.

So, common graphs can be interpreted as restricted property graphs: no labels on nodes, single labels on edges, no parallel edges with the same label, and no properties on edges. All these restrictions are direct consequences of the nature of the RDF data model.

While these restrictions seem severe at a first glance, the resulting data model can actually easily simulate unrestricted property graphs: labels on nodes can be simulated with the presence of corresponding keys, edges can be materialised as nodes if we need properties over edges or parallel edges with the same label. This means not only that common graphs can be used without loss of generality in expressiveness and complexity studies, but also that the corresponding restricted property graphs are flexible enough to be usable in practice, while additionally guaranteeing interoperability with the RDF data model.

## A.3 Class information

The common graph data model does not have direct support for class information. The reason for this is that RDF and property graphs handle class information rather differently. In RDF, both class and instance information is part of the graph data itself: classes are elements of the graph, subclass-superclass relationships are represented as edges between classes, and membership relationships are represented as edges between elements and classes. In property graphs, membership of a node in a class is indicated by a label put on the node. A node can belong to many classes, but the only way to say that class $A$ is a subclass of class $B$ is to ensure in the schema that each node with label $A$ also has label $B$. That is,

- in property graphs class membership information is available locally in a node, but consistency must be ensured by the schema,
- in RDF, obtaining class membership information requires navigating in the graph, but consistency is for free.

Clearly, both approaches have their merits, but when passing from one to the other data needs to be translated. This means that we cannot pick one of these approaches for the common data model while keeping it a natural submodel of both RDF and property graphs. Therefore, to reduce the complexity of this study, we do not include in our common data model any dedicated features for supporting class information. Note, however, that common graphs can support both these approaches indirectly: designated predicates can be used to represent membership and subclass relationships, and keys with a dummy value can simulate node labels.

## B STANDARD SHACL

Standard SHACL defines shapes as a conjunction of *constraint components*. The different constructs from our formalization correspond to fundamental building blocks of these constraint components. Next to that, the formalization of SHACL presented in this paper is less expressive than standard SHACL: both because we defined it here for the common data model (which is a strict subset of RDF), and because we want to simplify our narrative. For example, we leave out the comparison of RDF terms using sh:lessThan to simplify our story. Furthermore, because the common graph data-model does not define language tags, the corresponding constraint components from standard SHACL are left out.

Our formalization is closely tied to the ones found in the literature. There, correspondence between the formalization of the literature and standard SHACL has been shown in detail [26]. This section highlights and discusses some relevant details.

*Class targets and constraint component.* As a consequence of the common graph data model not directly supporting the modelling of classes, some class-based features are not adapted in our formalization. Specifically, there are no selectors ("target declarations" in standard SHACL) that involve classes. Furthermore, the value type constraint component sh:class is not covered by our formalization.

*Closedness.* In standard SHACL syntax, closedness is a property that takes a true or false value. The semantics of closedness is based on a list of predicates that are allowed for a given focus node. This list can be inferred based on the predicated used in property shapes, or this list can be explicitly given using the sh:ignoredProperties keyword. Our formalization effectively adopts the latter approach: $\text{closed}(Q)$ means that the properties mentioned in the set $Q$ are the ignored properties.

*Path expressions.* The path expressions used in our formalization deviate from the standard in three obvious ways. First, we make a distinction between 'keys' and 'predicates'. This is simply a consequence of using our common graph data model. Second, we leave out some of the immediately available path constructs from standard SHACL: one-or-more paths and zero-or-one path. However, these are expressible using the building blocks of our formalization: one-or-more paths are expressed as $\pi \cdot \pi^*$, and zero-or-one paths are expressed as $\pi \cup \text{id}$. Lastly, our path expression allow for writing the identity relation explicitly. This cannot be done in literal standard SHACL syntax, but its addition to the formalization serves to highlight its hidden presence in the language. Writing the identity relation directly in a counting construct, e.g., $\exists^{\geq n}\text{id}.\top$, never adds expressive power. In the case of $n = 1$, the shape is always satisfied (and thus equivalent to $\top$), and it is easy to see that for any $n > 1$, it is never satisfied. The situation with complex path expressions in counting constructs is less clear from the outset. However, it has been shown [7] (Lemma 3.3), that the only case where id adds expressiveness is with complex path expressions of the form $\pi \cup \text{id}$. This is exactly the definition of zero-or-one paths and is therefore covered by standard SHACL. Another place where id can occur in our formalization is in the equality and disjointness constraints, e.g., $\text{eq}(\text{id}, p)$. According to the standard SHACL recommendation,

you cannot write this shape. However, in the SHACL Test Suite [21] test core/node/equals-001, the following shape is tested for:

```
ex:TestShape
  rdf:type sh:NodeShape ;
  sh:equals ex:property ;
  sh:targetNode ex:ValidResource1 .
```

on the following data:

```
ex:ValidResource1
  ex:property ex:ValidResource1 .
```

The intended meaning of this test is, in natural language: "The targetnode ex:ValidResource1 has a ex:property self-loop and no other ex:property properties". Effectively, this is the semantics for our eq(id, $p$) construct. The situation with disj(id, $p$) is similar.

We therefore have an ambiguous situation: the standard description of SHACL does not allow for shapes of the form eq(id, $p$), but the test suite, and therefore all implementations that pass it completely, do[3]. It then seems fair to include this powerful construct in the formalization.

*Comparisons with constants.* A direct consequence of the assumption that node identities in the common graph data model are hidden from the user, our abstraction of SHACL on common graphs does not support comparisons with constants from $\mathcal{N}$. Comparisons with constants from $\mathcal{V}$ are allowed.

*Node tests.* Our formalization uses the test($\mathbb{v}$) construct to denote many of the node tests available in SHACL. We list the tests from standard SHACL that are covered by this construct.

**DatatypeConstraintComponent** Tests whether a node has a certain datatype.

**MinExclusiveConstraintComponent**
**MinInclusiveConstraintComponent**
**MaxExclusiveConstraintComponent**
**MaxInclusiveConstraintComponent** These four constraints cover can check whether a node is larger (**Max**) or smaller (**Min**) than some value, and whether this forms a partial order (**Inclusive**) or a strict, or total, order (**Exclusive**). Based on the SPARQL < or ≤ operator mapping.

**MaxLengthConstraintComponent**
**MinLengthConstraintComponent** These two constraints test whether the length of the lexical form of the node is "larger" or equal (resp. "smaller" or equal) than some provided integer value. Strictly speaking, the recommendation defines these constraint components also on IRIs. However, we limit their use to Literals.

**PatternConstraintComponent** Tests whether the length of the lexical form of the node satisfies some regular expression. Strictly speaking, the recommendation defines these constraint components also on IRIs. However, we limit their use to Literals.

Then there are two types of tests not covered by our formalization:

---
[3]Incidentally, all implementations currently mentioned in the implementation report handle these cases correctly.

**NodeKindConstraintComponent** Tests whether a node is an IRI, Blank Node, or Literal. Our tests apply only to RDF Literals.

**LanguageInConstraintComponent** Test whether the language tag of the node is one of the specified language tags. This feature is not supported by our data model, since it lacks language tags.

## C STANDARD SHEX

The Shape Expressions Language (ShEx) [42] and the ShapeMaps language [41] are defined by the Shape Expressions Community group[4] at W3C. Hereafter we use *standard ShEx* to refer to the language defined in [42], while *ShEx* designates the language presented in this paper.

In this appendix we support the following

**Claim 1.** *On common graphs, the expressive power of ShEx schemas is equivalent to the expressive power of standard ShEx schemas without recursion.*

Section C.3 explains standard ShEx on common graphs, while Section C.2 explains standard ShEx without recursion.

### C.1 Standard ShEx schema and the validation problem

A standard ShEx schema is a set of named shape expressions, and is usually formalised as a pair $(L, def)$ where $L$ is a finite set of shape names (in practice these are IRIs), and $def$ is a function that associates a shape expression with every shape name. In standard ShEx, the validation problem $\mathcal{G} \vDash \mathcal{S}$ from Section 2.4 is not defined as such. That is, the ShEx specification [42] does not say what it means for a graph to be valid w.r.t. a standard ShEx schema; it only defines what it means for a node in a graph to satisfy a shape expression.

However, the problem considered in practice is whether some selected nodes in the graph satisfy some indicated shape expressions from the schema. This is specified by a shape map [41]. A shape map can be formalised as a set of pairs of the form $(sel, l)$, where $l \in L$ and $sel$ is a unary query. While the shape map specification [41] allows the selectors from Definition 7, most implementations allow general SPARQL queeries as selectors.

In the current paper, we integrate the shape map into the schema itself, which allows us to specify the validation problem in a uniform way for the three graph schema formalisms considered. Additionally, in shape maps we do not use shape names but shape expressions directly; the next section argues why this is not a problem from the point of view of expressive power.

### C.2 Shape names and recursion

Recursion is an important mechanism in standard ShEx. In the current paper however we consider only ShEx schemas without recursion (to be defined shortly). This restriction was made because neither standard SHACL[5] nor PG-Schema allow for recursion.

---
[4]https://www.w3.org/community/shex/
[5]There exist formalizations of SHACL that introduce recursion, but recursion is not specified in the standard.

In standard ShEx, the fact that shape expressions are named allows to refer to them using their name. In particular, references allow for circular recursive definitions. As an example, consider the standard ShEx schema on Figure 2. It contains the single shape name `ex:User` whose definition is given by the shape expression inside the curly braces. The latter shape expression refers to itself: `@ex:User` indicates a reference to the shape expression named `ex:User`. Concretely, the shape expression requires from an RDF node to have an `ex:email` predicate whose value is a string, as well as any number of `ex:invited` predicates whose values are nodes that satisfy the shape expression named `ex:User`.

```
PREFIX ex: <http://ex.example/#>
PREFIX xsd: <http://www.w3.org/2001/XMLSchema#>
ex:User {
  ex:email   xsd:string ;
  ex:invited @ex:User *
}
```

**Figure 2: A standard ShEx schema.**

A standard ShEx schema $(L, def)$ is considered with recursion when there is a shape name $l \in L$ whose definition uses a reference $@l$ to itself, either directly or transitively through references to other shape names. Every standard ShEx schema without recursion can be rewritten to an equivalent schema without references, simply by replacing every reference by its definition. In other words, references in standard ShEx do not add expressive power for non-recursive schemas.

## C.3 Syntax of standard ShEx on common graphs

The syntax and semantics of standard ShEx differ slightly from the ones presented here. There are some purely syntactic differences, as for instance the boolean operators that are called and, or and not in standard ShEx, for which we adopt here the usual mathematical notation $\wedge$, $\vee$ and $\neg$. Another difference concerns the so-called node constraints of standard ShEx. These are constraints to be verified on the actual node of an RDF graph (which is an IRI, a literal or a blank node) without considering its neighbourhood. As pointed out in Section A.1, node constraints for nodes (i.e. elements of $\mathcal{N}$) are irrelevant for common graphs. Standard ShEx node constraints on values correspond to the atomic shape expressions $\text{test}(\mathbb{v})$ and $\text{is}(c)$ in ShEx. Their expressive power can be entirely captured by selecting for $\mathcal{T}$ a language equivlent to node constraints on values in standard ShEx. Note finally that the $\text{test}(\mathbb{any})$ in ShEx allows to distinguish nodes from values.

The grammar presented on Figure 3 gives an abstract syntax for standard ShEx *shape expressions se* and *triple expressions te* for common graphs. Note that we use the ShEx syntax for the boolean operators and for node constraints. The other differences are:

- In standard ShEx, the atomic shape expression that defines the neighbourhood of a node (non-terminal *sh*) is parameterised by a set of extra (possibly inversed) predicates and keys $Q$, indicated by the keyword extra. In Section C.4.1 we show that extra is syntactic sugar in standard ShEx.
- In standard ShEx, the atomic shape expression derived from the non-terminal *sh* can have an optional closed modifier.

On the other hand, ShEx introduces the triple expressions $\neg P$ and $\neg P^-$. As we will see, the latter are used when translating standard ShEx to ShEx in order to distinguish between closed and non-closed standard ShEx shape expressions.

- Triple expressions in ShEx contain the atomic expression $\varepsilon$, while standard ShEx does not allow it directly. On the other hand standard ShEx allows to use intervals of the form $[min; max]$ to define bounded or unbounded repetition, while ShEx allows only the unbounded repetition $*$. We show in Section C.4.2 that the two variants have equivalent expressive power.
- Triple constraints in standard ShEx (rule *tc*) allow to use a . (dot) instead of the shape expression, which is in fact equivalent to the ShEx shape expression $\{\top\}$.

Note that, strictly speaking, the extra predicates and keys are optional in standard ShEx. However, an absent extra set is equivalent to extra $\emptyset$, therefore we will consider that it is always present.

$$se ::= \text{is}(c) \mid \text{test}(\mathbb{v}) \mid sh \mid \text{closed } sh \mid se \wedge se \mid se \vee se \mid \neg se .$$

$$sh ::= \text{extra } Q \ \{te\} .$$

$$te ::= tc \mid te ; te \mid te|te \mid te[min; max] .$$

$$tc ::= q \ se \mid q \ . .$$

$$\text{with } c \in \mathcal{N} \cup \mathcal{V}, \mathbb{v} \in \mathcal{T}, q \in \mathcal{P} \cup \mathcal{K} \cup \mathcal{P}^- \cup \mathcal{K}^-,$$
$$Q \subseteq_{fin} \mathcal{P} \cup \mathcal{K} \cup \mathcal{P}^- \cup \mathcal{K}^-, min \in \mathbb{N} \text{ and } max \in \mathbb{N} \cup \{*\}.$$

**Figure 3: Abstract syntax for standard ShEx.**

## C.4 Translations between ShEx and standard ShEx

In this section we introduce a back and forth translation between standard ShEx without recursion and ShEx. We claim that these translations preserve the semantics w.r.t. the validity of a graph. The claim is presented without proof as it would require to define formal semantics for standard ShEx. The proof is however not difficult to make using the formal semantics from [8].

*C.4.1 Eliminating* extra *from standard ShEx.* We show on an example how the extra constructed can be eliminated in standard ShEx. The same example will be used later on for the translation from standard ShEx to ShEx, therefore it is described in detail.

*Example 13.* Consider the standard ShEx shape expression

$$se = \text{extra } \{p_1, p_2\} \ \{te\}$$
$$\text{with} \qquad te = p_1 \{p .\} ; p_1 \{p' .\} ; p_3 . ; p_4^- .$$
$$\text{and} \qquad p_1, p_2, p_3, p_4, p, p' \in \mathcal{P} \cup \mathcal{K}$$

that has a set of extra predicates and keys $\{p_1, p_2\}$. It is satisfied by nodes whose neighbourhood has the following outgoing triples:

(1) one $p_1$-triple leading to a node that satisfies $\{p .\}$,
(2) another $p_1$-triple leading to a node that satisfies $\{p' .\}$,
(3) a $p_3$-triple leading to an unconstrained node,

(4) because $p_1$ appears as extra, other $p_1$-triples are also allowed as long as they satisfy **none** of the constraints present for $p_1$ in $te$, that is, they satisfy neither $\{p \;.\}$ nor $\{p' \;.\}$,

(5) because $p_2$ appears as extra, $p_2$-triples are allowed and their target is not constrained because $p_2$ does not appear in the triple expression $te$,

(6) finally, because the shape is not closed, all outgoing triples whose predicate is not $p_1$ neither $p_3$ are allowed, $\{p_1, p_3\}$ being the set of non-inversed predicates that occur directly in $te$. Formally, $\{p_1, p_3\} = preds(te) \cap \mathcal{P} \cap \mathcal{K}$, see Section C.4.3.

The node has also the following incoming triples:

(7) one incoming $p_4$-triple coming from an unconstrained node,

(8) because standard ShEx does not close the incoming triples, all incoming triples whose predicate is different from $p_4$ are allowed.

The shape expression $se$ from Example 13 is equivalent to the following shape expression without extra:

$$\left\{ te \;;\; te_{p_1}^* \;;\; te_{p_2}^* \right\}$$

where

$$te_{p_1} = p_1 \; (\neg \{p \;.\} \land \neg \{p' \;.\}) \qquad \text{and} \qquad te_{p_2} = (p_2 \;.)$$

The sub-expression $te_{p_1}^*$ allows to satisfy the requirement (4) from Example 13, while the sub-expression $te_{p_2}^*$ allows to satisfy requirement (5).

This construction for eliminating extra can be generalised to arbitrary shape expressions. The idea is to combine (with the ; operator) the initial triple expression with a sub-expression of the form $q \; se_q^*$ for every (possibly inverse) extra predicate $q$, where $se_q$ is the conjunction of the negated shape expressions $se'$ such that $q \; se'$ directly in $te$.

Without loss of generality, **from now on, we consider only standard ShEx shape expressions without** extra.

*C.4.2    Normalised triple expressions.* We now show how standard ShEx triple expressions and triple expressions can be normalised so that they use a limited number of operators, which will be useful for the translation between standard ShEx and ShEx.

*Normalisation of standard ShEx triple expressions.* A standard ShEx triple expression is normalised when it uses only the intervals $[0; 1]$ and $[0; *]$. Standard ShEx triple expressions are normalised using rewriting based on the following equivalences:

$$te[min; *] = te[0; *] \;;\; \underbrace{te \;;\; \cdots \;;\; te}_{min \text{ times}}$$

$$te[min; max] = \underbrace{te \;;\; \cdots \;;\; te}_{min \text{ times}} \;;\; \underbrace{te[0; 1] \;;\; \cdots \;;\; te[0; 1]}_{max-min \text{ times}} \qquad \text{when } max \neq *$$

*Normalisation of ShEx triple expressions.* Here after, $f$ designates a ShEx triple expression derivable from the third rule of the grammar in Definition 6. For every ShEx triple expression $f$, we define $f^? = f \mid \varepsilon$. A ShEx triple expression $f$ is normalised if either $f = \varepsilon$, or $f$ does not use $\varepsilon$ as sub-expression, but can use the ? operator defined here-above. Every triple expression can be normalised by

eliminating occurrences of $\varepsilon$ using these two properties and the ? operator:

- it is a neutral element for the ; operator, that is, $f \;;\; \varepsilon = \varepsilon \;;\; f = f$ for every ShEx triple expression $f$,
- $\varepsilon^* = \varepsilon$.

Without loss of generality, **from now on we consider only normalised triple expressions**.

*C.4.3    Direct predicates of triple expressions.* This section is devoted to two technical definitions. For every triple expression we define the set of (possibly inversed) predicates and keys that appear directly in the expression. Formally, if $f$ is a ShEx triple expression derived by the third rule of the grammar in Definition 6, then we define the set $preds(f) \subseteq \mathcal{P} \cup \mathcal{K} \cup \mathcal{P}^- \cup \mathcal{K}^-$ inductively on the structure of $f$ by:

$$preds(\varepsilon) = \emptyset$$
$$preds(p.\varphi) = \{p\}$$
$$preds(p^-.\varphi) = \{p^-\}$$
$$preds(\varphi \;;\; \varphi') = preds(\varphi) \cup preds(\varphi')$$
$$preds(\varphi \mid \varphi') = preds(\varphi) \cup preds(\varphi')$$
$$preds(\varphi^*) = preds(\varphi)$$

For a standard ShEx triple expression $te$, the set $preds(te)$ is defined similarly (recall that $q \in \mathcal{P} \cup \mathcal{K} \cup \mathcal{P}^- \cup \mathcal{K}^-$):

$$preds(q \; se) = \{q\}$$
$$preds(q \;.) = \{q\}$$
$$preds(se \;;\; se') = preds(se) \cup preds(se')$$
$$preds(se \mid se') = preds(se) \cup preds(se')$$
$$preds(te[min; max]) = preds(se)$$

*C.4.4    Translation from standard ShEx to ShEx.* With every standard ShEx shape expression $se$ we associate the ShEx shape expression $\tau(se)$ as defined in Table 9. It is defined by mutual recursion with the corresponding translation function $\tau_e(te)$ for standard ShEx triple expressions $te$ presented in Table 8.

**Table 8: Translation from standard ShEx to ShEx for normalised triple expressions, with $q \in \mathcal{P} \cup \mathcal{K} \cup \mathcal{P}^- \cup \mathcal{K}^-$.**

| $te$ | $\tau_e(te)$ |
| --- | --- |
| $q \; se$ | $p. \tau(se)$ |
| $q \;.$ | $p. \{\top\}$ |
| $te \;;\; te'$ | $\tau_e(te) \;;\; \tau_e(te')$ |
| $te \mid te'$ | $\tau_e(te) \mid \tau_e(te')$ |
| $te[0; *]$ | $\tau_e(te)^*$ |
| $te[0; 1]$ | $\tau_e(te) \mid \varepsilon$ |

*C.4.5    Translation from ShEx to standard ShEx.* Unless otherwise specified, in the sequel $f$ designates a ShEx triple expression produced by the third rule in the grammar in Definition 6. In Table 10 we present a function that with every normalised ShEx triple expression $f$ associates the standard ShEx triple expression $\sigma_e(f)$. It is defined by mutual recursion with the translation function that

**Table 9: Translation from standard ShEx to ShEx for shape expressions.**

| $se$ | $\tau(se)$ |
|------|------------|
| $\mathrm{is}(c)$ | $\mathrm{is}(c)$ |
| $\tau(\mathrm{test}(\mathbb{v}))$ | $\mathrm{test}(\mathbb{v})$ |
| $se \wedge se'$ | $\tau(se) \wedge \tau(se')$ |
| $se \vee se'$ | $\tau(se) \vee \tau(se')$ |
| $\neg se$ | $\neg \tau(se)$ |
| closed $\{te\}$ | $\tau_{\mathrm{e}}(te) ;(\neg Q^-)^*$ |
| | with $Q = preds(te) \cap (\mathcal{P}^- \cup \mathcal{K}^-)$ |
| $\{te\}$ | $\tau_{\mathrm{e}}(te) ;(\neg Q^-)^* ;(\neg P)^*$ |
| | with $Q = preds(te) \cap (\mathcal{P}^- \cup \mathcal{K}^-)$ |
| | and $P = preds(te) \cap (\mathcal{P} \cup \mathcal{K})$ |

with every ShEx shape expression $\varphi$ associates a standard ShEx shape expression $\sigma(\varphi)$, and that will be presented shortly. Note that the case $f = \varepsilon$ is omitted in Table 10: recall that in normalised ShEx triple expressions, $\varepsilon$ can only appear standalone (not in subexpressions), therefore this case $f = \varepsilon$ will be treated with shape expressions.

**Table 10: Translation from ShEx to standard ShEx for normalised triple expressions.**

| $f$ | $\sigma_{\mathrm{e}}(f)$ |
|-----|-------------------------|
| $p.\varphi$ | $p \; \sigma_{\mathrm{e}}(\varphi)$ |
| $p^-.\varphi$ | $p^- \; \sigma_{\mathrm{e}}(\varphi)$ |
| $f ; f'$ | $\sigma_{\mathrm{e}}(f) ; \sigma_{\mathrm{e}}(f')$ |
| $f \mid f'$ | $\sigma_{\mathrm{e}}(f) \mid \sigma_{\mathrm{e}}(f')$ |
| $f^*$ | $\sigma_{\mathrm{e}}(f)[0; *]$ |
| $f^?$ | $\sigma_{\mathrm{e}}(f)[0; 1]$ |

The definition of $\sigma(\varphi)$ is straightforward for the following cases:

$$\sigma(\mathrm{is}(c)) = \mathrm{is}(c)$$
$$\sigma(\mathrm{test}(\mathbb{v})) = \mathrm{test}(\mathbb{v})$$
$$\sigma(\varphi \wedge \varphi') = \sigma(\varphi) \wedge \sigma(\varphi')$$
$$\sigma(\varphi \vee \varphi') = \sigma(\varphi) \vee \sigma(\varphi')$$
$$\sigma(\neg\varphi) = \neg\sigma(\varphi)$$

The remaining case is for a shape expression of the form $\{e\} = \{f ; \cdots\}$ where $f$ is normalised. Consider the most general case

$$e = f ;(\neg P^-)^* ;(\neg Q)^*$$

Let also

$$\{p_1, \ldots, p_m\} = (preds(f) \cap \mathcal{P}^- \cap \mathcal{K}^-) \setminus P$$
$$\{q_1, \ldots, q_n\} = (preds(f) \cap \mathcal{P} \cap \mathcal{K}) \setminus Q.$$

Intuitively, $\{p_1, \ldots, p_m\}$ is the set of predicates that are not allowed to appear on incoming edges in the neighbourhoods defined by $e$, and similarly $\{q_1, \ldots, q_n\}$ are the forbidden predicates for outgoing

edges. Then

$$\sigma(\{e\}) = \left\{ \begin{array}{l} \sigma_{\mathrm{e}}(f) \; ; \\ p_1^- \,.[0;0] ; \cdots ; p_m^- \,.[0;0] \; ; \\ q_1 \,.[0;0] ; \cdots q_n \,.[0;0] \end{array} \right\}$$

If $f = \varepsilon$, then the term $\sigma_{\mathrm{e}}(f)$ on the first line of the definition of $\sigma(\{e\})$ must be omitted.

The remaining case for the definition of $\sigma(\{e\})$ is for

$$e = f ;(\neg P^-)^*$$

Let $\{p_1, \ldots, p_m\}$ be as before. Then

$$\sigma(\{e\}) = \mathrm{closed} \left\{ \begin{array}{l} \sigma_{\mathrm{e}}(f) \; ; \\ p_1^- \,.[0;0] ; \cdots ; p_m^- \,.[0;0] \end{array} \right\}$$

As before, if $f = \varepsilon$, then the term $\sigma_{\mathrm{e}}(f)$ must be omitted.

# D STANDARD PG-SCHEMA

We present here a version of PG-Schema that is restricted to common graphs, and therefore as such a simplified version of the original PG-Schema, but stays closer to the original form of PG-Schema as discussed in [2] than the version in the body of this paper. It is meant to illustrate how the version of PG-Schema in the body describes indeed a reasonable core of the original full PG-Schema.

The central idea of standard PG-Schema and of PG-Schema in [2] is that a schema (called *graph type* in this context) consists of three parts: (1) a set of node types, (2) a set of edge types, and (3) a set of graph constraints that represents logical statements about the property graph that must hold for it to be valid. A particular property graph is then said to be valid wrt. such a graph type if (1) every node in the property graph is in the semantics of at least one node type, (2) every edge in the property graph is in the semantics of at least one edge type, and (3) the property graph satisfies all specified graph constraints.

The organisation of this section is as follows. We first discussion the notions of *node types* and *edge types*. After that we discuss how path expressions are defined, after which we discuss what graph constraints look like in this setting. In the final two subsections we discuss how this version of PG-Schema relates the original defined in [2], and how ot relates to the one define in this paper.

## D.1 Node types

The purpose of node types in PG-Schema is to describe nodes, their properties and their labels. Since in the common graph model nodes no longer have labels, node types become simply record types where the record fields describe the allowed keys. In addition it is indicated with these record type whether they are *closed* or *open*, where the first indicates that only the indicated keys are allowed, and the later that additional keys are allowed. This leads to the following formal definition.

DEFINITION 15 (NODE TYPE). *A node type is an expression $\mathbb{n}$ of the form defined by the grammar*

$$\mathbb{n} ::= \{\} \;\big|\; \{\}^\circ \;\big|\; \{k : \mathbb{v}\} \;\big|\; \{k : \mathbb{v}\}^\circ \;\big|\; \mathbb{n} \,\&\, \mathbb{n} \;\big|\; \mathbb{n} \mid \mathbb{n} \,.$$

*where $k \in \mathcal{K}$ and $\mathbb{v} \in \mathcal{T}$.*

Here $\{\}^\circ$ indicates the open record type without any keys, $\{k : \mathbb{v}\}^\circ$ is the open record type with just key $k$ of type $\mathbb{v}$.

**Table 11: Semantics of standard PG-path expressions.**

| $\pi$ | $[\![\pi]\!]^{\mathcal{G}} \subseteq (\mathcal{N} \cup \mathcal{V}) \times (\mathcal{N} \cup \mathcal{V})$ for $\mathcal{G} = (E, \rho)$ |
|---|---|
| $\mathbb{e}$ | $\{(u, v) \mid \exists p : (u, p, v) \in E \land (\rho(u), p, \rho(v)) \in [\![\mathbb{e}]\!]\}$ |
| $\neg \mathbb{e}$ | $\{(u, v) \mid \exists p : (u, p, v) \in E \land (\rho(u), p, \rho(v)) \notin [\![\mathbb{e}]\!]\}$ |

For these node types $\mathbb{n}$ we define a value semantics $[\![\mathbb{n}]\!]$, just like for content types, which is defined in the same way except that $[\![\{\}^{\circ}]\!] = [\![\top]\!]$ and $[\![\{k : \mathbb{v}\}^{\circ}]\!] = [\![\{k : \mathbb{v}\} \,\&\, \top]\!]$. It follows straightforwardly that for each content type there is an equivalent node type and vice versa.

## D.2 Edge types

In PG-Schema there is a notion of edge type, which consists of three parts: (1) a type describing the source node, (2) a type describing describing the contents of the edge itself, and (3) a type describing the the target node. Since in common graphs the content of an edge is just a label, a type describing this content can be simply an expression of the form $\star$ (a wild-card indicating that any label is possible) or a finite set $P$ of labels (indicating that only these labels are allowed). So we get the following definition for edge types.

DEFINITION 16 (EDGE TYPE). *An* edge type *is an expression $\mathbb{e}$ of the form defined by the grammar*

$$\mathbb{e} ::= \mathbb{n} \xrightarrow{\star} \mathbb{n} \mid \mathbb{n} \xrightarrow{P} \mathbb{n} \mid \mathbb{e} \,\&\, \mathbb{e} \mid \mathbb{e} \mid \mathbb{e}.$$

where $P$ is a finite subset of $\mathcal{P}$.

As for node types, we define for edge types a value semantics, which in this case defines which combinations of (1) source node content, (2) edge content, and (3) target node content are allowed.

DEFINITION 17 (VALUE SEMANTICS OF EDGE TYPES). *With an edge type $\mathbb{e}$ we associate a* value semantics $[\![\mathbb{e}]\!] \subseteq \mathcal{R} \times \mathcal{P} \times \mathcal{R}$ *which is defined with induction on the structure of $\mathbb{e}$ as follows:*

(1) $[\![\mathbb{n}_1 \xrightarrow{\star} \mathbb{n}_2]\!] = [\![\mathbb{n}_1]\!] \times \mathcal{P} \times [\![\mathbb{n}_2]\!]$

(2) $[\![\mathbb{n}_1 \xrightarrow{P} \mathbb{n}_2]\!] = [\![\mathbb{n}_1]\!] \times P \times [\![\mathbb{n}_2]\!]$

(3) $[\![\mathbb{e}_1 \,\&\, \mathbb{e}_2]\!] = \{ ((r_1 \cup s_1), p, (r_2 \cup s_2)) \in \mathcal{R} \times \mathcal{P} \times \mathcal{R} \mid (r_1, p, r_2) \in [\![\mathbb{e}_1]\!] \land (s_1, p, s_2) \in [\![\mathbb{e}_2]\!]\}$

(4) $[\![\mathbb{e}_1 \mid \mathbb{e}_2]\!] = [\![\mathbb{e}_1]\!] \cup [\![\mathbb{e}_2]\!]$

## D.3 Path expressions

We define here a notion of path expression that we call *standard PG-path expression* and that is similar to the notion of PG-path expression of Definition 9, except that in the positions where a content type $\mathbb{c}$ is allowed, we allow the use of a node type $\mathbb{n}$ or edge type $\mathbb{e}$. As discussed earlier, node types are equivalent to content types, and so we can given them the same semantics in path expressions. Edge types, however, are different, and their semantics in path expressions is given in Table 11.

## D.4 Graph constraints

The graph constraints in PG-Schema are based on the constraints discussed in PG-Keys [3]. Although the latter paper focuses on key constraints, it also discusses other closely related cardinality constraints. We capture these constraints here in the context of the common graph data model with the following formal definition.

DEFINITION 18 (PG-CONSTRAINT). *A* PG-constraint *is a formula of one of the following three forms:*

**K1:** $\forall x : \varphi(x) \Leftarrow \mathbf{Key}\; \bar{y} : \psi(x, \bar{y})$
**K2:** $\forall x : \varphi(x) \rightarrow \exists^{\leq k}\; \bar{y} : \psi(x, \bar{y})$
**K3:** $\forall x : \varphi(x) \rightarrow \exists^{\geq k}\; \bar{y} : \psi(x, \bar{y})$

*where $x$ is a variable that ranges over nodes and values, $\varphi(x)$ and $\psi(x, \bar{y})$ are formulas of the form $\exists \bar{z} : \xi$ with $\bar{z}$ a vector of node and value variables and $\xi$ a conjunction of atoms of the form $\pi(z_i, z_j)$ with $z_i$ and $z_j$ either equal to $x$, in $\bar{y}$, or in $\bar{z}$, and $\pi$ a standard PG-path expression, such that the free variables in $\varphi(x)$ are just $x$ and those in $\psi(x, \bar{y})$ are $x$ and the variables in $\bar{y}$.*

The intuition of the constraints of the form **K1** is that they define a key constraint for all values or nodes selected by $\varphi(x)$. It states for such $x$ that any vector of values and/or nodes $\bar{y}$ that satisfies $\psi(x, \bar{y})$ identifies $x$, i.e., is associated with at most one such $x$. So the symbol $\Leftarrow$ should be read here as stating that the left-hand side is functionally determined by the right-hand side. More formally, its semantics is defined as being equivalent with the formula $\forall \bar{y} : \exists^{\leq 1} x : \varphi(x) \land \psi(x, \bar{y})$.

For the constraints of the forms **K2** and **K3** the interpretation is simply the usual one in first-order logic.

## D.5 The relationship with PG-Schema on full Property Graphs

The presented definitions introduce two important simplification w.r.t./ the original in [2]: (1) It is defined over common graphs which simplifies the property graph data model in several ways and (2) it assumes what is called the STRICT semantics of a graph type in [2] and ignores the LOOSE semantics. We briefly discuss here why these simplification preserve the essential characteristics of the original schema language.

*D.5.1 Concerning the simplification of the data model.* Recall that common graphs simplify property graphs in three ways: (1) nodes only have properties and no labels, (2) edges only have one label and no properties, and (3) edges have no independent identity. However, these features can be readily simulated in the common graph data model. For example. edges with identity can be simulated by nodes that have an outgoing edge with label *source* and an outgoing edge with label *target* to respectively the source node and the target node of the simulated edge. Moreover, labels can be simulated by introducing a special dummy value $\Lambda$ that is used for keys that represent labels. For example, a node $n$ where $\rho(n)$ contains the pairs (*Person*, $\Lambda$), (*Employee*, $\Lambda$), (*hiringDate*, *12-Dec-2023*), and (*fulltime*, *true*), simulates a node with labels *Person* and *Employee*, and properties *hiringDate* and *fulltime*.

It is not hard to see how under such a simulation the PG-Schema presented in this section could simulate a more powerful schema language where we could use tests in path expressions for the presence (or absence) of (combinations of) labels in path expressions and tests for presence (or absence) of (combinations of ) properties of edges. Moreover, we could navigating via a simulated an edge and test for certain properties with a path expression of the form

$source^- \cdot \pi \cdot target$ where $\pi$ simulates any test over the content of the edge. Finally, we could straightforwardly simulate key and cardinality constraints for edges.

*D.5.2 Concerning the STRICT and LOOSE semantics.* In PG-Schema as described in [2] there is a separate LOOSE semantics defined for graph types. In that case the set of nodes types and the set of edge types in the graph type are ignored and a property graph is said to be already valid wrt. a graph type if it satisfies all graph constraints in the graph type. Note that in the original PG-Schema this did not mean that the sets of nodes types and edge types are entirely ignored, since there they also have the function of defining short-hands that can be used in the definitions of the graph constraints.

The LOOSE interpretation can be easily simulated in the version presented here by letting the set of node types consist of $\top$, the trivial node type, and the set of edge types consist of $\top \xrightarrow{\star} \top$, the trivial edge type.

## D.6 The relationship with PG-Schema in the body of the paper

The constraints of the forms **K2** and **K3** are very similar to the selector-shape pairs presented for PG-Schema in Section 5. Indeed, the selector is in this case the formula $\varphi(x)$ and the shape are the formulas of the forms $\exists^{\leq k} \bar{y} : \psi(x, \bar{y})$ and $\exists^{\geq k} \bar{y} : \psi(x, \bar{y})$. However, there are also several notable differences: (1) The schema only consists of constraints and does not separately mention node or edge types. (2) There are no edge types in path expressions. (3) All constraints are restricted so that $\bar{y}$ is just a single variable. (4) There are no constraints of the form **K1**. (5) The constraints are syntactically restricted such that $\varphi(x)$ is restricted to just one atom, so the form $\exists z : \pi(x, z)$, and $\psi(x, \bar{y})$ is restricted to just one atom, so the form $\pi(x, y)$. Apart from these restriction, there is also a generalisation, since in Section 5 the shape expression are closed under intersection. That this does not change the expressive power is easy to see, since a selector-shape pair of the form $(sel, (\varphi_1 \wedge \varphi_2))$ can always be replaced with the combination of the pairs $(sel, \varphi_1)$ and $(sel, \varphi_2)$ without changing the semantics of the schema.

In the following subsections we discuss the previously mentioned restrictions.

*D.6.1 No separate sets of node types and edge types.* It is not hard to show that this can be simulated. Assume for example we have a graph type with a set of node types $\{\mathbb{n}_1, \mathbb{n}_2, \mathbb{n}_3\}$. The check that each node must be in the semantics of at least one of these node types can be simulated by the constraint

$$\forall x : \top(x, x) \to \exists y : (\mathbb{n}_1 \mid \mathbb{n}_2 \mid \mathbb{n}_3)(x, y)$$

Since node types are closed under the $\mid$ operator, $(\mathbb{n}_1 \mid \mathbb{n}_2 \mid \mathbb{n}_3)$ is a node type, and therefore an allowed path expression. Recall that a node type acts in a path expression as the identity relation restricted to nodes that are in the semantics of that type.

Similarly, if the set of edge types of a graph type is $\{\mathbb{e}_1, \mathbb{e}_2, \mathbb{e}_3\}$, we can ensure that each edge is in the semantics of at least one of these edge types using the constraint

$$\forall x : \top(x, x) \to \exists^{\leq 0} y : \neg(\mathbb{e}_1 \mid \mathbb{e}_2 \mid \mathbb{e}_3)(x, y).$$

*D.6.2 No edge types in path expressions.* It is not hard to show that path expressions that contains tests involving edge types can be rewritten to equivalent path expressions that do not use edge types.

We first consider the non-negated edge types in path expressions. We start with the observation that we can normalise edge types to a union of edge types that do not contain the $\mid$ operator. This is based on the following equivalences for path semantics that allow us to push down the $\mid$ operator:

- $(\mathbb{m}_1 \mid \mathbb{m}_2) \xrightarrow{\alpha} \mathbb{m}_3 \equiv (\mathbb{m}_1 \xrightarrow{\alpha} \mathbb{m}_3 \mid \mathbb{m}_2 \xrightarrow{\alpha} \mathbb{m}_3)$
- $\mathbb{m}_1 \xrightarrow{\alpha} (\mathbb{m}_2 \mid \mathbb{m}_3) \equiv (\mathbb{m}_1 \xrightarrow{\alpha} \mathbb{m}_2 \mid \mathbb{m}_1 \xrightarrow{\alpha} \mathbb{m}_3)$

In a next normalisation step we can remove bottom-up the & operator using the following rules, where we use the symbol $\mathbb{e}_\emptyset$ to denote the empty edge type:

- $(\mathbb{m}_1 \xrightarrow{\alpha} \mathbb{m}_2) \,\&\, (\mathbb{m}_3 \xrightarrow{\beta} \mathbb{m}_4) \equiv (\mathbb{m}_1 \,\&\, \mathbb{m}_3) \xrightarrow{\alpha \sqcap \beta} (\mathbb{m}_3 \,\&\, \mathbb{m}_4)$

where $\sqcap$ is defined such that (1) $\star \sqcap P = P \sqcap \star = P$ for $P \subseteq \mathcal{P}$, and (2) $P \sqcap Q = P \cap Q$ for $P, Q \subseteq \mathcal{P}$.

As a final normalisation step we get rid of edge types $\mathbb{m}_1 \xrightarrow{P} \mathbb{m}_2$ where $P$ contains two or more predicates, by applying the rule:

- $\mathbb{m}_1 \xrightarrow{\{p_1, \ldots, p_k\}} \mathbb{m}_2 \equiv (\mathbb{m}_1 \xrightarrow{\{p_1\}} \mathbb{m}_2 \mid \ldots \mid \mathbb{m}_1 \xrightarrow{\{p_k\}} \mathbb{m}_2)$

After these normalisation steps we will have rewritten the edge type to the form $(\mathbb{e}_1 \mid \ldots \mid \mathbb{e}_k)$ with each $etype_i$ of one of the following forms: (1) $\mathbb{m}_1 \xrightarrow{\star} \mathbb{m}_2$, (2) $\mathbb{m}_1 \xrightarrow{\{p\}} \mathbb{m}_2$, and (3) $\mathbb{m}_1 \xrightarrow{\emptyset} \mathbb{m}_2$. We can express such an edge type $(\mathbb{e}_1 \mid \ldots \mid \mathbb{e}_k)$ as a path expression $(\pi_1 \cup \ldots \cup \pi_k)$, where each $\pi_i$ is constructed as follows:

- $\mathbb{m}_1 \xrightarrow{\star} \mathbb{m}_2 \equiv \mathbb{m}_1 \cdot \neg\emptyset \cdot \mathbb{m}_2$
- $\mathbb{m}_1 \xrightarrow{\{p\}} \mathbb{m}_2 \equiv \mathbb{m}_1 \cdot p \cdot \mathbb{m}_2$
- $\mathbb{m}_1 \xrightarrow{\emptyset} \mathbb{m}_2 \equiv \neg\top$

Recall that $\neg\top$ is the negation of the trivial node type and so in a path expression represents the empty binary relation.

We now turn our attention to negated edge types. These are normalised in the same as we described before, and we end up with an edge type of the form $\neg(\mathbb{e}_1 \mid \ldots \mid \mathbb{e}_k)$ with each $\mathbb{e}_i$ a primitive edge type. This can also be represented as a union of path expressions $(\pi_1 \cup \ldots \cup \pi_m)$ where each $\pi_j$ is a path expression the expresses one way that the edge does not conform to any of the types in $\mathbb{e}_1, \ldots, \mathbb{e}_k$. To illustrate this consider as an example the following negated edge type:

$$\neg\big(\mathbb{m}_1 \xrightarrow{\{p\}} \mathbb{m}_2 \mid \mathbb{m}_1' \xrightarrow{\{p'\}} \mathbb{m}_2'\big)$$

This can be simulated in a path expression by replacing it with the following path expression:

$$
\begin{aligned}
&\neg\mathbb{m}_1 \cdot \neg\mathbb{m}_1' \cdot \neg\emptyset \ \cup \ \neg\mathbb{m}_1 \cdot \neg\{p'\} \ \cup \ \neg\mathbb{m}_1 \cdot \neg\emptyset \cdot \neg\mathbb{m}_2' \ \cup \\
\cup\ &\neg\mathbb{m}_1' \cdot \neg\{p\} \ \cup \ \neg\{p, p'\} \ \cup \ \neg\{p\} \cdot \neg\mathbb{m}_2' \ \cup \\
\cup\ &\neg\mathbb{m}_1' \cdot \neg\emptyset \cdot \neg\mathbb{m}_2 \ \cup \ \neg\{p'\} \cdot \neg\mathbb{m}_2 \ \cup \ \neg\emptyset \cdot \neg\mathbb{m}_2 \cdot \neg\mathbb{m}_2'
\end{aligned}
$$

Note that this indeed enumerates all the ways that an edge could not be in the semantics of $\big(\mathbb{m}_1 \xrightarrow{\{p\}} \mathbb{m}_2 \mid \mathbb{m}_1' \xrightarrow{\{p'\}} \mathbb{m}_2'\big)$. Basically we pick for each of the primitive edge types whether the edge is not in the semantics because of (1) the source node, (2) the label, or (3) the target node.

*D.6.3 Only single variable counting.* The restriction to allow only one variable in $\bar{y}$ is introduced because in SHACL and ShEx all the counting is also restricted to single values and nodes, rather than tuples of values and nodes. Although this is often useful in real-world data modelling, e.g., to represent composite keys, this restriction is introduced to make PG-Schema more comparable to SHACL and ShEx.

*D.6.4 No **K1** constraints.* After restricting to single-variable counting, **K1** constraints are of the form

$$\forall x : \varphi(x) \Leftarrow \textbf{Key } y : \psi(x, y) .$$

Their semantics is defined by the formula $\forall \bar{y} : \exists^{\leq 1} x : \varphi(x) \wedge \psi(x, \bar{y})$.

If $y$ matches nodes (which can be detected based on path expressions used in the atoms involving $y$), we can equivalently express this constraint as

$$\forall y : \top(y, y) \rightarrow \exists^{\leq 1} x : \varphi(x) \wedge \psi(x, y)$$

If $y$ matches values, then it is used in the first position of an atom whose path expressions begins from $k^-$ or in the second position of an atom whose path expression ends with $k$. In either case, we can equivalently express this constraint as

$$\forall y : (k^- \cdot k)(y', y) \rightarrow \exists^{\leq 1} x : \varphi(x) \wedge \psi(x, y)$$

*D.6.5 Only one atom in formulas.* This restriction of the query language underlying PG-Schema limits the expressive power of PG-Schema, but similar restrictions would be imposed anyway on the Common Graph Schema Language by the limitations of SHACL and ShEx. Some additional expressive power could be gained by allowing tree-shaped conjunctions of atoms with at most 2 free variables, but this would further complicate the formal development.

## E MORE ON THE CORE

In this section we prove Proposition 1. Recall that common shapes are defined by the grammar

$$\varphi ::= \exists \pi \mid \exists^{\leq n} \pi_1 \mid \exists^{\geq n} \pi_1 \mid \exists \mathbb{c} \wedge \nexists \neg P \mid \varphi \wedge \varphi .$$

$$\mathbb{c} ::= \{\} \mid \{k : \mathbb{v}\} \mid \mathbb{c} \& \mathbb{c} \mid \mathbb{c} \mid \mathbb{c} .$$

$$\pi_0 ::= \{k : c\} \mid \neg\{k : c\} \mid \mathbb{c} \& \top \mid \neg(\mathbb{c} \& \top) \mid \pi_0 \cdot \pi_0 .$$

$$\pi_1 ::= \pi_0 \cdot p \cdot \pi_0 \mid \pi_0 \cdot p^- \cdot \pi_0 \mid \pi_0 \cdot k \mid k^- \cdot \pi_0 .$$

$$\bar{\pi} ::= \pi_0 \mid p \mid \bar{\pi}^- \mid \bar{\pi} \cdot \bar{\pi} \mid \bar{\pi} \cup \bar{\pi} .$$

$$\pi ::= \bar{\pi} \mid \bar{\pi} \cdot k \mid k^- \cdot \bar{\pi} \mid k^- \cdot \bar{\pi} \cdot k' .$$

where $n \in \mathbb{N}$, $P \subseteq_{fin} \mathcal{P}$, $k, k' \in \mathcal{K}$, $c \in \mathcal{V}$, and $p \in \mathcal{P}$. We will refer to PG-path expressions defined by the nonterminal $\pi_0$ in the grammar as *filters*.

The following two subsections describe the translations of common schemas to SHACL and ShEx. The translations are very similar but we include them both for the convenience of the reader.

## E.1 Translation to SHACL

**Lemma 1.** *For each open content type $\mathbb{c}$ there is a SHACL shape $\varphi_{\mathbb{c}}$ such that $\mathcal{G}, v \vDash \varphi_{\mathbb{c}}$ iff $\rho(v) \in [\![\mathbb{c}]\!]$ for all $\mathcal{G} = (E, \rho)$ and $v \in \text{Nodes}(\mathcal{G})$.*

PROOF. For the content type $\top$ the corresponding SHACL shape is $\top$. For a content type of the form

$$\{k_1 : \mathbb{v}_1\} \& \{k_2 : \mathbb{v}_2\} \& \cdots \& \{k_m : \mathbb{v}_m\} \& \top ,$$

the corresponding SHACL shape is

$$\exists k_1.\text{test}(\mathbb{v}_1) \wedge \exists k_2.\text{test}(\mathbb{v}_2) \wedge \cdots \wedge \exists k_m.\text{test}(\mathbb{v}_m) .$$

Finally, every open content type different from $\top$ can be expressed as

$$(\mathbb{c}_1 \mid \cdots \mid \mathbb{c}_\ell) \& \top ,$$

where each $\mathbb{c}_i$ is a content type of the form $\{k_1 : \mathbb{v}_1\} \& \{k_2 : \mathbb{v}_2\} \& \cdots \& \{k_m : \mathbb{v}_m\}$ for some $m$. The corresponding SHACL shape is

$$\varphi_1 \vee \cdots \vee \varphi_\ell ,$$

where $\varphi_i$ is the SHACL shape corresponding to the content type $\mathbb{c}_i \& \top$. □

**Lemma 2.** *For each filter $\pi_0$ there is a SHACL shape $\varphi_{\pi_0}$ such that $\mathcal{G}, v \vDash \varphi_{\pi_0}$ iff $(v, v) \in [\![\pi_0]\!]^{\mathcal{G}}$ for all $\mathcal{G}$ and $v \in \text{Nodes}(\mathcal{G})$.*

PROOF. By Lemma 1, the claim holds for $\pi_0 = \mathbb{c} \& \top$. For $\{k : c\}$ the corresponding SHACL shape is $\exists k.\text{is}(w)$. As SHACL shapes are closed under negation, the claim holds for $\neg\{k : c\}$ and $\neg(\mathbb{c} \& \top)$. Finally, concatenations of filters correspond to conjunctions of shapes, so the claim follows because SHACL shapes are closed under conjunction. □

**Lemma 3.** *For each common shape of the form $\exists \pi$ there is a SHACL shape $\varphi_{\exists \pi}$ such that $\mathcal{G}, v \vDash \varphi_{\exists \pi}$ iff $\mathcal{G}, v \vDash \exists \pi$ for all $\mathcal{G}$ and $v \in \text{Nodes}(\mathcal{G}) \cup \text{Values}(\mathcal{G})$.*

PROOF. Let us first look at common shapes of the form $\exists \pi$ where $\pi$ is a concatenation of filters and atomic path expressions of the form $p$, $p^-$, $k$, or $k^-$. Without loss of generality we can assume that the concatenation ends with a filter or with $k$. We proceed by induction on the length of the concatenation. The base cases are $\exists \pi_0$ and $\exists k$, which correspond to $\varphi_{\pi_0}$ (Lemma 2) and $\exists k.\top$. For $\exists \pi_0 \cdot \pi$ we can take $\varphi_{\pi_0} \wedge \varphi_{\exists \pi}$. For $\exists p \cdot \pi$ we can take $\exists p.\varphi_{\exists \pi}$, and similarly for $\exists p^- \cdot \pi$ and $\exists k^- \cdot \pi$.

The general case follows because SHACL shapes are closed under union. Indeed, because our PG-path expressions are star-free, we can assume without loss of generality that in each common shape of the form $\exists \pi$, the PG-path expression $\bar{\pi}$ underlying $\pi$ is a union of concatenations of filters and atomic path expressions of the form $p$ or $p^-$. Then, for

$$\exists k^- \cdot (\pi^1 \cup \cdots \cup \pi^m) \cdot k'$$

we can take

$$\varphi_{\exists k^- \cdot \pi^1 \cdot k'} \vee \cdots \vee \varphi_{\exists k^- \cdot \pi^m \cdot k} .$$

Simiarly for $\exists k^- \cdot (\pi^1 \cup \cdots \cup \pi^m)$, $\exists (\pi^1 \cup \cdots \cup \pi^m) \cdot k'$, and $\exists (\pi^1 \cup \cdots \cup \pi^m)$. □

**Lemma 4.** *For each common shape $\varphi$ there is a SHACL shape $\hat{\varphi}$ such that $\mathcal{G}, v \vDash \varphi$ iff $\mathcal{G}, v \vDash \hat{\varphi}$ for all $\mathcal{G}$ and $v \in \text{Nodes}(\mathcal{G}) \cup \text{Values}(\mathcal{G})$.*

PROOF. Because SHACL shapes are closed under conjunction, it suffices to prove the claim for the atomic common shapes of the forms $\exists \pi$, $\exists^{\leq n} \pi_1$, $\exists^{\geq n} \pi_1$, and $\exists \mathbb{c} \wedge \nexists \neg P$. The first case follows from Lemma 3.

Let us look at common shapes of the form $\exists^{\geq n}\pi_1$. If $n = 0$ we can simply take $\top$. Suppose $n > 0$. Then, for

$$\exists^{\geq n}\pi_0 \cdot p \cdot \pi_0'$$

we can take

$$\varphi_{\pi_0} \wedge \exists^{\geq n}p.\varphi_{\pi_0'},$$

and similarly for $\exists^{\geq n}\pi_0 \cdot p^- \cdot \pi_0'$, $\exists^{\geq n}\pi_0 \cdot k^- \cdot \pi_0'$, and $\exists^{\geq n}\pi_0 \cdot k$ (using $\top$ instead of $\varphi_{\pi_0'}$).

Next, we consider common shapes of the form $\exists^{\leq n}\pi_1$. For

$$\exists^{\leq n}\pi_0 \cdot p \cdot \pi_0'$$

we can take

$$\neg\varphi_{\pi_0} \vee \exists^{\leq n}p.\varphi_{\pi_0'},$$

and similarly for $\exists^{\leq n}\pi_0 \cdot p^- \cdot \pi_0'$, $\exists^{\leq n}\pi_0 \cdot k^- \cdot \pi_0'$, and $\exists^{\leq n}\pi_0 \cdot k$ (again, using $\top$ instead of $\varphi_{\pi_0'}$).

Finally, let us consider a common shape of the form $\exists\,\mathbb{c} \wedge \nexists\neg P$. Suppose first that

$$\mathbb{c} = \{\}.$$

Then, the corresponding SHACL shape is simply

$$\text{closed}(P).$$

Next, suppose that

$$\mathbb{c} = \{k_1 : \mathbb{v}_1\} \& \ldots \& \{k_m : \mathbb{v}_m\}.$$

Then, the corresponding SHACL shape is

$$\exists k_1.\text{test}(\mathbb{v}_1) \wedge \cdots \wedge \exists k_m.\text{test}(\mathbb{v}_m) \wedge \text{closed}\big(\{k_1, \ldots, k_m\} \cup P\big).$$

In general, as in Lemma 1, we can assume that

$$\mathbb{c} = \mathbb{c}_1 \mid \ldots \mid \mathbb{c}_m$$

where each $\mathbb{c}_i$ is of one of the two forms considered above. The corresponding SHACL shape is then

$$\varphi_1 \vee \cdots \vee \varphi_m$$

where $\varphi_i$ is the SHACL shape corresponding to $\exists\mathbb{c}_i \wedge \nexists\neg P$, obtained as described above. □

**Lemma 5.** *For every common schema there is an equivalent SHACL schema.*

Proof. Let $\mathcal{S}$ be a common schema. We obtain an equivalent SHACL schema $\mathcal{S}'$ by translating each $(sel, \varphi) \in \mathcal{S}$ to $(sel', \varphi')$ such that for all $\mathcal{G}$ and $v \in \mathcal{N} \cup \mathcal{V}$,

$$\mathcal{G}, v \vDash sel \text{ implies } \mathcal{G}, v \vDash \varphi$$
$$\text{iff}$$
$$\mathcal{G}, v \vDash sel' \text{ implies } \mathcal{G}, v \vDash \varphi'.$$

Recall that $sel$ is a common shape of one of the following forms:

$$\exists k,\ \exists p \cdot \pi,\ \exists p^- \cdot \pi,\ \exists\{k : c\} \cdot \pi,\ \exists\big(\{k : \mathbb{v}\} \& \top\big) \cdot \pi,\ \exists k^- \cdot \pi.$$

For $sel'$ we take, respectively,

$$\exists k.\top,\quad \exists p.\top,\quad \exists p^-.\top,\quad \exists k.\top,\quad \exists k.\top,\quad \exists k^-.\top.$$

For $\varphi'$ we take $\neg\varphi_{sel} \vee \hat{\varphi}$ where $\varphi_{sel}$ is obtained using Lemma 3 and $\hat{\varphi}$ is obtained using Lemma 4. □

## E.2 Translation to ShEx

**Lemma 6.** *For each open content type $\mathbb{c}$ there is a ShEx shape $\varphi_{\mathbb{c}}$ such that $\mathcal{G}, v \vDash \varphi_{\mathbb{c}}$ iff $\rho(v) \in [\![\mathbb{c}]\!]$ for all $\mathcal{G} = (E, \rho)$ and $v \in \text{Nodes}(\mathcal{G})$.*

Proof. For the content type $\top$ the corresponding ShEx shape is $\{\top\}$.

For a content type of the form

$$\{k_1 : \mathbb{v}_1\} \& \cdots \& \{k_m : \mathbb{v}_m\} \& \top,$$

the corresponding ShEx shape is

$$\{k_1.\text{test}(\mathbb{v}_1)\,;\top\} \wedge \cdots \wedge \{k_m.\text{test}(\mathbb{v}_m)\,;\top\}.$$

Finally, every other open content type can be expressed as

$$(\mathbb{c}_1 \mid \cdots \mid \mathbb{c}_\ell) \& \top,$$

where each $\mathbb{c}_i$ has the form $\{k_1 : \mathbb{v}_1\} \& \cdots \& \{k_m : \mathbb{v}_m\}$ for some $m$. The corresponding ShEx shape is

$$\varphi_1 \vee \cdots \vee \varphi_\ell,$$

where $\varphi_i$ is the ShEx shape corresponding to the content type $\mathbb{c}_i \& \top$. □

**Lemma 7.** *For each filter $\pi_0$ there is a ShEx shape $\varphi_{\pi_0}$ such that $\mathcal{G}, v \vDash \varphi_{\pi_0}$ iff $(v, v) \in [\![\pi_0]\!]^{\mathcal{G}}$ for all $\mathcal{G}$ and $v \in \text{Nodes}(\mathcal{G})$.*

Proof. By Lemma 6, the claim holds for $\pi_0 = \mathbb{c} \& \top$. For $\{k : c\}$ the corresponding ShEx shape is $\{k.\text{is}(w)\}$. Because ShEx shapes are closed under negation, the claim also holds for $\neg\{k : c\}$ and $\neg(\mathbb{c} \& \top)$. Finally, concatenations of filters correspond to conjunctions of shapes, so the claim follows because ShEx shapes are closed under conjunction. □

**Lemma 8.** *For each common shape of the form $\exists\pi$ there is a ShEx shape $\varphi_{\exists\pi}$ such that $\mathcal{G}, v \vDash \varphi_{\exists\pi}$ iff $\mathcal{G}, v \vDash \exists\pi$ for all $\mathcal{G}$ and $v \in \text{Nodes}(\mathcal{G}) \cup \text{Values}(\mathcal{G})$.*

Proof. Let us first look at common shapes of the form $\exists\pi$ where $\pi$ is a concatenation of filters and atomic path expressions of the form $p$, $p^-$, $k$, or $k^-$. Without loss of generality we can assume that the concatenation ends with a filter or with $k$. We proceed by induction on the length of the concatenation. The base cases are $\exists\pi_0$ and $\exists k$, which correspond to $\varphi_{\pi_0}$ (Lemma 7) and $\{k.\{\top\}\,;\top\}$, respectively. For $\exists\pi_0 \cdot \pi$ we can take $\varphi_{\pi_0} \wedge \varphi_{\exists\pi}$. For $\exists p \cdot \pi$ we can take $\{p.\varphi_{\exists\pi}\,;\top\}$, and similarly for $\exists p^- \cdot \pi$ and $\exists k^- \cdot \pi$.

The general case follows because ShEx shapes are closed under union. Indeed, because our PG-path expressions are star-free, we can assume without loss of generality that in each common shape of the form $\exists\pi$, the PG-path expression $\bar{\pi}$ underlying $\pi$ is a union of concatenations of filters and atomic path expressions of the form $p$ or $p^-$. Then, for

$$\exists k^- \cdot (\pi^1 \cup \cdots \cup \pi^m) \cdot k'$$

we can take

$$\varphi_{\exists k^- \cdot \pi^1 \cdot k'} \vee \cdots \vee \varphi_{\exists k^- \cdot \pi^m \cdot k}.$$

Similarly for $\exists k^- \cdot (\pi^1 \cup \cdots \cup \pi^m)$, $\exists(\pi^1 \cup \cdots \cup \pi^m) \cdot k'$, and $\exists(\pi^1 \cup \cdots \cup \pi^m)$. □

**Lemma 9.** *For each common shape $\varphi$ there is a ShEx shape $\hat{\varphi}$ such that $\mathcal{G}, v \vDash \varphi$ iff $\mathcal{G}, v \vDash \hat{\varphi}$ for all $\mathcal{G}$ and $v \in \text{Nodes}(\mathcal{G}) \cup \text{Values}(\mathcal{G})$.*

Proof. Because ShEx shapes are closed under conjunction, it suffices to prove the claim for the atomic common shapes of the forms $\exists\pi$, $\exists^{\leq n}\pi_1$, $\exists^{\geq n}\pi_1$, and $\exists\mathbb{c}\wedge\not\exists\neg P$. The first case follows from Lemma 8.

Let us look at common shapes of the form $\exists^{\geq n}\pi_1$. If $n=0$ we can simply take $\{\top\}$. Suppose $n>0$. Then, for

$$\exists^{\geq n}\pi_0\cdot p\cdot\pi_0'$$

we can take

$$\varphi_{\pi_0}\wedge\left\{(p.\varphi_{\pi_0'})^n\,;\top\right\},$$

and similarly for $\exists^{\geq n}\pi_0\cdot p^-\cdot\pi_0'$, $\exists^{\geq n}\pi_0\cdot k^-\cdot\pi_0'$, and $\exists^{\geq n}\pi_0\cdot k$ (using $\{\top\}$ instead of $\varphi_{\pi_0'}$).

Next, we consider common shapes of the form $\exists^{\leq n}\pi_1$. For

$$\exists^{\leq n}\pi_0\cdot p\cdot\pi_0'$$

we can take

$$\neg\varphi_{\pi_0}\vee\neg\left\{(p.\varphi_{\pi_0'})^{n+1}\,;\top\right\}$$

and similarly for $\exists^{\leq n}\pi_0\cdot p^-\cdot\pi_0'$, $\exists^{\leq n}\pi_0\cdot k^-\cdot\pi_0'$, and $\exists^{\leq n}\pi_0\cdot k$ (again, using $\{\top\}$ instead of $\varphi_{\pi_0'}$).

Before we move on, let us introduce a bit of syntactic sugar. For a set $Q=\{q_1,q_2,\ldots,q_n\}\subseteq\mathcal{P}\cup\mathcal{K}$ we write $Q^*$ for the triple expression $(q_1.\{\top\}\,|\,q_2.\{\top\}\,|\ldots|\,q_n.\{\top\})^*$.

We are now ready to consider a common shape of the form $\exists\mathbb{c}\wedge\not\exists\neg P$. Suppose first that

$$\mathbb{c}=\{\}.$$

Then, the corresponding ShEx shape is simply

$$\left\{P^*\,;\left(\neg\emptyset^-\right)^*\right\}.$$

Next, suppose that

$$\mathbb{c}=\{k_1:\mathbb{v}_1\}\,\&\ldots\&\,\{k_m:\mathbb{v}_m\}.$$

Then, the corresponding ShEx shape is

$$\varphi_{\mathbb{c}\&\top}\wedge\left\{\{k_1,\ldots,k_m\}^*\,;P^*\,;\left(\neg\emptyset^-\right)^*\right\},$$

where $\varphi_{\mathbb{c}\&\top}$ is obtained from Lemma 6. In general, as in Lemma 6, we can assume that

$$\mathbb{c}=\mathbb{c}_1\,|\,\ldots\,|\,\mathbb{c}_m$$

where each $\mathbb{c}_i$ is of one of the two forms considered above. The corresponding ShEx shape is then

$$\varphi_1\vee\cdots\vee\varphi_m$$

where $\varphi_i$ is the ShEx shape corresponding to $\exists\mathbb{c}_i\wedge\not\exists\neg P$, obtained as described above. □

**Lemma 10.** *For every common schema there is an equivalent ShEx schema.*

Proof. Let $\mathcal{S}$ be a common schema. We obtain an equivalent ShEx schema $\mathcal{S}'$ by translating each $(sel,\varphi)\in\mathcal{S}$ to $(sel',\varphi')$ such that for all $\mathcal{G}$ and $v\in\mathcal{N}\cup\mathcal{V}$,

$$\mathcal{G},v\vDash sel\text{ implies }\mathcal{G},v\vDash\varphi$$
$$\text{iff}$$
$$\mathcal{G},v\vDash sel'\text{ implies }\mathcal{G},v\vDash\varphi'.$$

Recall that $sel$ is a common shape of one of the following forms:

$$\exists k,\ \exists p\cdot\pi,\ \exists p^-\cdot\pi,\ \exists\{k:c\}\cdot\pi,\ \exists\left(\{k:\mathbb{v}\}\,\&\,\top\right)\cdot\pi,\ \exists k^-\cdot\pi.$$

If $sel$ is of the form

$$\exists k,\quad\exists\{k:c\}\cdot\pi,\quad\text{or}\quad\exists\left(\{k:\mathbb{v}\}\,\&\,\top\right)\cdot\pi,$$

for $sel'$ we take $\{k.\{\top\}\,;\top\}$. In the remaining cases, we take, respectively,

$$\{p.\{\top\}\,;\top\},\quad\{p^-.\{\top\}\,;\top\},\quad\{k^-.\{\top\}\,;\top\}.$$

For $\varphi'$ we take $\neg\varphi_{sel}\vee\hat\varphi$ where $\varphi_{sel}$ is obtained using Lemma 8 and $\hat\varphi$ is obtained using Lemma 9. □

