# OpenReview forum: "Common Foundations for SHACL, ShEx, and PG-Schema"
_ACM.org/TheWebConf/2025/Conference — WWW 2025 Oral_

### Official Review · Reviewer_eLGn · 2024-11-29

**Novelty:** 5
**Technical Quality:** 6

**Review:**

The authors of the paper present an abstraction called Common Graphs, which aims to be an intermediate model between RDF graphs and (a restricted version of) Property Graphs. The authors then describe SHACL, (a fragment of) ShEx, and PG-schema over these common graphs. The paper ends by distilling a proposal for a schema language over Common Graphs that brings together common expressions from the three studied languages.

The paper is quite interesting and presents exhaustive exemplification to make the (at times rather complex) theory to a concrete level.

I have, however, several issues with the paper as it is. The first and most obvious one is that the appendices are longer than the paper itself. These appendices address the limitations of the common graphs to simulate property graphs, the actual mathematical development and justification of the abstractions of the schema languages, the reasoning behind why Shex's recursivity is not supported, etc. These details are not minor, and I would argue that they are fundamental to the understanding and utility of the paper. In this context, I question if a conference is the proper venue for this paper.

A small nitpick is that a common graph cannot simulate edge-less property-less nodes in property graphs. These do not exist in RDF, as nodes only occur in subjects and objects, so they must be part of an edge. In property graphs, however, isolated nodes can exist in a graph only by virtue of their ID, and common graphs cannot simulate this. I would also argue that these nodes that may denote the existence of something are not too uncommon.

Third, I would expect some empirical validation for a research paper. Has CoGSL been implemented? How can it be implemented? Does it have any empirical advantage or disadvantage over SHACL, SHEX, or PG-Schema? How verbose can be the validation rules? How can CoGSL, both theoretically and practically, be used for interoperability goals? How can the shapes be translated from one language to another via CoGSL? These technical, empirical aspects are fundamental to the impact of the paper, at least for the web conference.

Finally, a minor point is that the authors several times introduce notation, proceed to find it too verbose, and then use an abbreviated version. I understand that the theory of the paper can get too hard to follow, but these definitions-then-changes are not too helpful either.

**Questions:**

Why do you think that the contents of the appendices belong there? I would argue these are fundamental to understand the paper, and that the paper is not self-contained.

Are edge-less, property-less nodes uncommon, or rather, unimportant?

Why is there no empirical validation of the language? What are the concrete use cases that you envision? How can CoGSL be used as an intermediate layer for shape translation?

**Reviewer Confidence:**

3: The reviewer is confident but not certain that the evaluation is correct

**Scope:**

3: The work is somewhat relevant to the Web and to the track, and is of narrow interest to a sub-community

---

### Official Review · Reviewer_kiKN · 2024-12-02

**Novelty:** 6
**Technical Quality:** 7

**Review:**

In this paper, the authors compare three major graph schema languages, i.e. PG-Schema (developed for the Property Graphs data model) and SCHACL and ShEx (developed for the RDF data model). Indeed, one of the issues the involved communities (Semantic Web and Database, primarily) have had is where to find appropriate guidelines that analyse the pros and cons of the various solutions to understand which tool to adopt. This paper provides both a general (in the main text) and a technical (in the appendix) account of the similarities and differences between these three languages and discusses lucidly all the various aspects characterising them by adopting a common language, i.e. the Common Graph Schema Language, to enable such a comparison.

The contribution is undoubtedly valuable and fits the needs of the communities and practitioners in adopting schema languages for validating graph databases very well. The text is easy to follow, even if technical expertise may be needed to understand some of the passages. The results provided can be used as starting points for further studies that concern, for instance, implementations for converting shapes and constraints across different languages.

The only issue I perceive is of a practical and pragmatic nature. Indeed, such comparisons at a theoretical level are essential. However, when choosing a tool, the specific implementations available also play a role. For instance, I may decide that ShEx is the best language to adopt in theory, considering its features, but do I have an effective and efficient validation library for the programming language I want to use for a particular application? Thus, adding a comparison across these three languages that also includes whether the implementations available (in different programming languages) implement the full specification and present certain issues with particular constraints would be an ideal addition to the content of the current paper.

Reviewer: Silvio Peroni (https://orcid.org/0000-0003-0530-4305)

**Questions:**

* What is the current adoption of the three languages?
* What are the available software libraries that allow one to use these languages?
* In which programming languages have these libraries been developed?
* What are the issues that such implementations have concerning the features of the three languages?

**Reviewer Confidence:**

3: The reviewer is confident but not certain that the evaluation is correct

**Scope:**

3: The work is somewhat relevant to the Web and to the track, and is of narrow interest to a sub-community

---

### Official Review · Reviewer_VQAz · 2024-12-02

**Novelty:** 5
**Technical Quality:** 6

**Review:**

Strength:
This paper introduces the Common Graph Schema Language (CoGSL), which integrates the core functionalities of SHACL, ShEx, and PG-Schema, providing a unified framework for understanding the relationships between the three languages. This unified framework makes it easier for users to grasp and compare the languages, while also promoting interoperability across them.

Weakness:
1. Related work:
   - As a submission for 2025, the author should consider more recent works. Currently, there are only 5 references from 2023 and 7 from 2024 .
   - The referenced work Labra-Gayo J E.[1] and MillenniumDB[2] are representative results with high citation in the field and notable experimental performance. The analysis of these work, as well as a comparison with the your work, is missing.

2. Lack of detailed explanation:
 - In Section 2.1, it is stated that "It is easy to see that a common graph is a special case of a property graph," but there is no explanation provided for what is meant by "special case.".
 - In Section 4, The article does not provide much explanation for only considering non-recursive ShEx. It may be helpful to include an explanation as to why non-recursive ShEx is chosen, and what potential issues might arise when using recursive ShEx in practical applications.

3. Variable naming needs to be consistent. In Section 2.1, it states, "Then, a common graph can be seen as a finite set G ⊆ E such that for each 𝑢 ∈ N and 𝑘 ∈ K, there is at most one 𝑣 ∈ V," while in Section 2.2, it states, "We call 𝜌 (𝑣) the content of node 𝑣 in G." The variable 𝑣 has different meanings in each context,.

4. The article imposes certain restrictions and simplifications on SHACL, ShEx, and PG-Schema, and it should clarify how much of the original functionality is still supported after these simplifications. This will help readers better understand the practical value of the simplified versions and how they can be applied in real-world scenarios.

REFERENCES

[1] Labra-Gayo J E. Extending shape expressions for different types of knowledge graphs[C]//1st Workshop on Data Quality meets Machine Learning and Knowledge Graphs, DQMLKG, part of Extended Semantic Web Conference. 2024.

[2] Vrgoč D, Rojas C, Angles R, et al. MillenniumDB: An open-source graph database system[J]. Data Intelligence, 2023, 5(3): 560-610.

**Questions:**

In Section 6, we see that the Common Graph Schema Language (CoGSL) is defined as a fragment of PG-Schema, although it integrates some functionalities of SHACL and ShEx. Based on this approach, would it be appropriate for the title to be "Common Foundations for SHACL, ShEx, and PG-Schema"?

**Reviewer Confidence:**

3: The reviewer is confident but not certain that the evaluation is correct

**Scope:**

3: The work is somewhat relevant to the Web and to the track, and is of narrow interest to a sub-community

---

### Official Review · Reviewer_Csdu · 2024-12-03

**Novelty:** 6
**Technical Quality:** 7

**Review:**

# Common Foundations for SHACL, ShEx, and PG-Schema

This paper presents a common data model for SHACL, ShEx and PG-Schema, which are constraint languages for graph data. As the authors say in the Introduction section, these languages came from different point of views ("Semantic Web Based" descriptive based and PG-Schema prescriptive based). The authors present very clearly the three languages and their capabilities and next define what the common elements of these three languages are.

Strong points:
1. A clear presentation of the problem: I agree that the three methods for representing constraints over the two graph data models may be problematic: which one to chose, what capabilities they offer. The authors present very well the problem
2. Very clear definition of the capabilities of each constraint language: the authors present very clear definitions of each constraint language. I think this this the strongest point of the paper since these definitions allow to build upon them more results.
3. Clear and complete State of the art

Weak points:
1. Limited contribution: the contribution is minimal, the authors enumerate the commonalities from each constraint language. That enumeration is presented in Definition 12 (Common Shape). The main result is presented in Proposition 1, which is very limited

**Questions:**

1. The authors claim that they want to investigate recursive ShEx and more expressive query languages for PG-Schema more deeply. Why the authors did not do that in the current work?

**Reviewer Confidence:**

4: The reviewer is certain that the evaluation is correct and very familiar with the relevant literature

**Scope:**

4: The work is relevant to the Web and to the track, and is of broad interest to the community

---

### Official Review · Reviewer_dqMF · 2024-12-03

**Novelty:** 6
**Technical Quality:** 6

**Review:**

The paper “Common Foundations for SHACL, ShEx, and PG-Schema” compares the graph schema languages SHACL, ShEx and PG-Schema to each other and provides a common formalisation of the joint features of these three frameworks.

The paper first introduces a generalised graph notation that makes it possible to express Property graphs and RDF graphs using the same notation. The notation covers many important features but leaves out class declarations as a feature which are difficult to compare between these two frameworks. Next, the languages SHACL, ShEx and PG Schema are introduced making use of the notation. In a last section, the authors define a core schema language which only covers the features present in all three languages.

The paper is well-structured and the topic is very interesting (it is one of the rare cases where the reviewer was waiting for such a paper). The formalisation is concise and easy to follow. Only the target selection could be more detailed. All definitions are presented with examples. The main result of the paper, the fact that each common schema can be equivalently represented in Shacl and in ShEx is backed up by a proof in the appendix. The related work is discussed in detail and I personally find the paper an interesting read.

The only possible criticism I could come up with is, that the relation of property graphs and RDF is not that well understood yet and that  maybe they are so different in nature that it does not make sense to compare schema languages for such different targets. But even if this comparison is not that relevant in practice, because the property graph user cannot really choose between SHACL and PG-Schema, it is still interesting to see how different communities tackle the problem of expressing graph schemata.

More detailed questions/comments:
- differences between the development of SHACL/ShEx and PG-Schema (page 1, line 78ff): the differences do not look that big to me. Maybe I am mistaken, but I think that the papers about PG-Schema do not fully define its semantics? The papers also mention implementations.
- common data model (page 2, line 143ff): why are nodes, values, predicates and keys disjoint? For mapping from and to RDF, it would make sense to not have the disjointness between, e.g., keys and predicates. From a purely RDF point of view, the same would hold for nodes and edges. But for the latter case I totally understand the choice given that the schema languages do not provide any support for, e.g., comparing predicates to nodes.
- related to the previous point, it might be worth mentioning that the authors assume two types of properties in RDF just as the datatype property and the object property in OWL-DL.
- shacl selector (page 3, Definition 4): It looks like the selector “is(c)” is meant to mimic a node target, therefore I am surprised that it is only defined for values and not for nodes. Is there a reason for that?
- selectors in general: I only understood from the context how the selectors work, it might be helpful to explain these better (can be very short)

**Questions:**

Questions can be found above in the detailed comments.

**Reviewer Confidence:**

3: The reviewer is confident but not certain that the evaluation is correct

**Scope:**

4: The work is relevant to the Web and to the track, and is of broad interest to the community